# The Robustness Limits of SoTA Vision Models to Natural Variation

**Mark Ibrahim**                                                    *marksibrahim@meta.com*

**Quentin Garrido**

**Ari Morcos**

**Diane Bouchacourt**

*Fundamental AI Research (FAIR), Meta*

**Reviewed on OpenReview:** *https://openreview.net/forum?id=QhHLwn3DOY*

## Abstract

Recent state-of-the-art vision models have introduced new architectures, learning paradigms, and larger pretraining data, leading to impressive performance on tasks such as classification. While previous generations of vision models were shown to lack robustness to factors such as pose, the extent to which this next generation of models are more robust remains unclear. To study this question, we develop a dataset of more than 7 million images with controlled changes in pose, position background, lighting color, and size. We study not only how robust recent state-of-the-art models are, but also the extent to which models can generalize to variation in each of these factors. We consider a catalog of recent vision models, including vision transformers (ViT), self-supervised models such as masked autoencoders (MAE), and models trained on larger datasets such as CLIP. We find that even today's best models are not robust to common changes in pose, size, and background. When some samples varied during training, we found models required a significant portion of instances seen varying to generalize—though eventually robustness did improve. When variability is only witnessed for some classes however, we found that models did not generalize to other classes unless the classes were very similar to those seen varying during training. We hope our work will shed further light on the blind spots of SoTA models and spur the development of more robust vision models.

## 1 Introduction

A dataset of natural images can be described by a set of factors of variations which characterize the main axes along which samples sample vary; for example pose, position, illumination, size, etc (Bengio et al., 2013; Bouchacourt et al., 2021). Importantly, test-time unseen data samples may

exhibit different variability across factors than those seen during training (Quinonero-Candela et al., 2009). It is thus desirable for state-of-the-art (SoTA) models to be robust to changes in these factors (Bengio et al., 2013). However, previous work has shown that vision models such as Convolutional Neural Networks (CNNs) or Vision Transformers (ViTs; Dosovitskiy et al. (2021)) are quite brittle to changes in pose, illumination, or even slight rotations and translation transformations (Engstrom et al., 2019; Alcorn et al., 2019; Abbas & Deny, 2022). Yet, much of the existing work focuses either on the effect of a single transformation or analyzes toy settings where variability can be controlled. If we aim to deploy models in more realistic and challenging applications, however, we need to study their brittleness to more natural variations on more realistic data which can potentially appear together (e.g. multiple factors at the same time).

Here, we extend existing work to study models' susceptibility to changes in position, size, spot hue, background, and pose independently, as well as *changes in all factors in conjunction*. To do so, we develop a dataset allowing based on 3d warehouse objects (Trimble Inc) that we place in non-uniform backgrounds and for which we vary the aforementioned factors. Using the typical evaluation procedures of self-supervised models (Caron et al., 2021; Dosovitskiy et al., 2021; Chen et al., 2020b) (finetuning and linear evaluation), we examine robustness across a catalog of state-of-the-art vision architectures, such as CLIP (Radford et al., 2021) that have significantly outperformed earlier models on robustness benchmarks such as ObjectNet (Barbu et al., 2019), Masked AutoEncoders (MAE, (He et al., 2022)), or ViTs Dosovitskiy et al. (2021)) among others. This allows us to compare common inductive biases such as architectures, training paradigm or the amount of pre-training data. Furthermore, we examine the effect of *variability* for the factors, that is (i) seeing some instances varying for a *single factor affects the other factors* and how (ii) seeing some *some classes varying for factors affect other classes*. To the best of our knowledge, generalization of robustness across classes has not previously been studied.

Our main findings and contributions, summarized in Figure 1, are:

1. We study the robustness of a wide range of SoTA models to variations in naturally occurring factors. We found **SoTA pre-trained models fine-tuned with little or no variability are not robust to factor variations** (Section 4).

2. We compare the effect of different inductive biases as realized through different architectures, training paradigms, quantity of pre-training data, and finetuning vs. linear evaluation. We found differences in **architecture and training paradigm have minor impacts on robustness, but that more training data helps** and that finetuning generally leads to worse robustness (Section 4).

3. **Increasing the amount of variability of all instances for each factor during training helps generalization** (Section 5.1). However, increasing variability only for some instances can hurt if not enough variability is introduced (Section 5.2). Nonetheless, **variability in single factors tends to improve robustness to other factors too** (Section 5.2).

4. By studying the effect of variability across classes, we find that if a **class is seen varying for some factors during training, it helps to generalize to very similar classes that were not encountered varying, but generalizes worse to all classes that are even little dissimilar** and much worse to those classes which are highly dissimilar (Section 5.4).

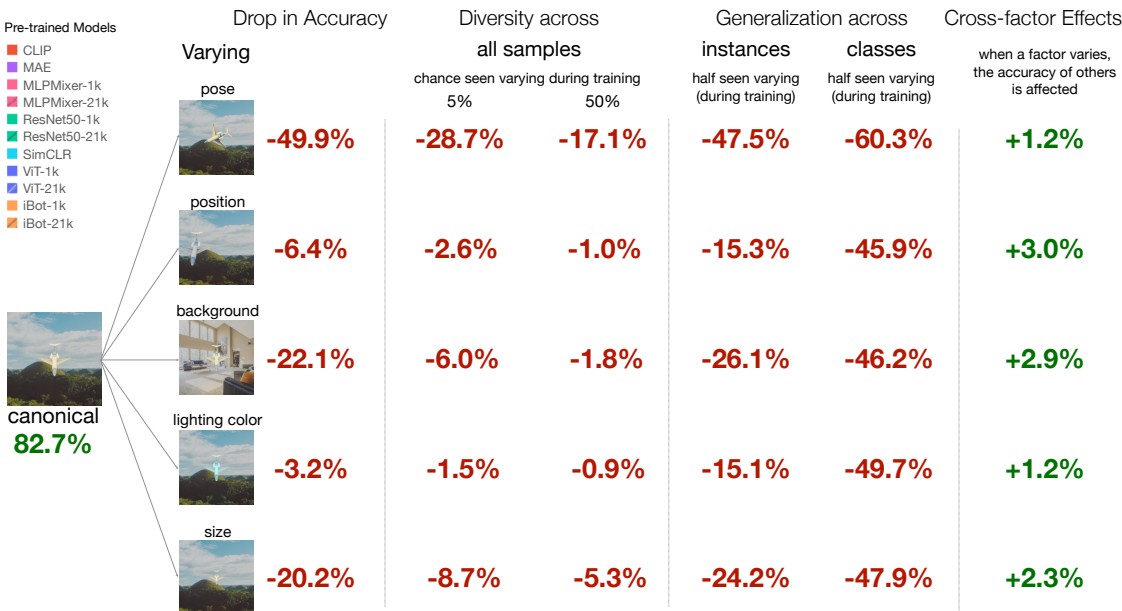

Figure 1: **SoTA models are not robust to and struggle to generalize common variations in pose, background, size**. We show the average drop in accuracy across models when we vary each factor. We find if we vary all samples during training, models require a significant portion of variation ($\geq 50\%$) to close the robustness gaps. When variation is only seen for some classes or instances, models struggle to generalize variation across instances or classes. Finally, when a factor varies during training the robustness of other factors is also affected.

## 2 Related work

Recently, there have been much interesting work studying models' brittleness. Our work falls in this body of literature, yet we aim to provide a more extensive analysis by (i) varying different factors alone and in combination (ii) studying the effect of different amounts of variability seen in the training data (iii) on a extended list of standard vision models. Alcorn et al. (2019) focuses on models' robustness to pose changes, while Engstrom et al. (2019) studies rotation and translation changes (both together and alone) and find that explicitly augmenting the data with such variations does not fix the problem, a conclusion shared by (Azulay & Weiss, 2019) who study small image changes e.g. translating / scaling. Madan et al. (2021) also find that ResNets and CLIP networks are brittle to pose and lighting changes. Madan et al. investigate the generalization of only CNNs to combinations of two factors (object category and 3D viewpoint), finding that increasing the number of combinations seen at training helps generalization and also that separate networks outperform shared ones.

Some works explicitly study the invariance of models, e.g. Bordes et al. (2021); Bouchacourt et al. (2021) where the latter also found that data augmentation does not bring the expected invariance. In (von Kügelgen et al., 2021) the preservation of natural transformation information was studied

in a self-supervised setting, where they found that the pretraining data augmentation policy plays an important role. Eulig et al. (2021) develops a benchmark for measuring generalization to six factors (position, scale, hue, lighting, shape, and texture) using a synthetic dataset of digits focusing on several (mostly CNN-based) models. Similar to us, Djolonga et al. (2020) studies robustness of models with respect to factors of variation of objects on various backgrounds, and how inductive biases such as the amount of data or size of model impact performance. However, they only focus on Convolutional Neural Networks (CNNs), while we also study vision transformers based architectures, feedforward models and even zero-shot multi-modal models such as CLIP. Furthermore, they factors in the limited setting where only 2D in-plan orientation changes are possible, while we use realistic 3D renderings of objects. More importantly, we extend our study to the case where variability in each factor is added *during training*, and not just at test-time. This allows us to gain an in-depth understanding of models ability to generalize changes seen during training to new objects, rather than out-of-the-box robustness. Perhaps closest to our work, (Abbas & Deny, 2022) studied the sensitivity of a large body of models and a variety of effects (architectures, data augmentation, dataset modalities), albeit only to pose changes (orientation and scale). In this work, we extend the set of factors studied, including cross-factors effects and consider a catalog of recent SoTA models spanning various learning paradigms, architectures, and sizes.

## 3 Methods

### 3.1 The dataset suited to perform robustness analysis

To study the brittleness of state-of-the-art models with respect to data factors of variation, several data properties are desirable. First, while there exist variants of real image datasets that present naturally occurring variations (Hendrycks et al., 2021), control over the data generation process allows detailed factor metadata. Second, we want the dataset to vary sufficiently for us to draw consistent conclusions. Finally, we want the images to remain as close to realistic images in terms of image quality as possible. Existing datasets developed to show robustness of models often vary in just one or a few factors, and the images are not really realistic or span only a few classes (e.g. Shapes3D (Kim & Mnih, 2018), MPI3D (Gondal et al., 2019), dSprites (Higgins et al., 2017) among others). Therefore we develop our own dataset based on 3d Warehouse (Trimble Inc) objects that we place in non-uniform backgrounds.

We use 54 synsets from 3d Warehouse (Trimble Inc), and 50 objects for each synset. For the first 4 scalar factors (position, pose, size, lighting color), we use equally spaced scalar values. For the background, we use 5 background types (*sky, water, city, home, grass*) and 5 different backgrounds per type, with natural images coming from Li et al. (2022; 2021b;a). We define for each factor a canonical value, that is, the most represented value for that factor, to mimic the fact that in natural images we often see objects in a set of given factors (e.g. their upright position). We then vary these factors in three different manners: (i) each factor independently (101 scalar values equally spaced for scalar factors + 25 backgrounds) (ii) factors varying in pairs (11 scalar values + 10 backgrounds) (iii) all factors varying together (drawing 1000 random combinations from the full grid of 11 equally spaced values and 10 backgrounds). This gives us roughly 7 million (M) images in total, divided as follows: single factor (1.1M), paired factors (3.1M), and all factors (2.7M). We provide more details on the generation process in Appendix Section B.

### 3.2 Introduce pre-trained models

We select a set of state-of-the-art (SoTA) vision models, with many achieving $> 80\%$ top-1 accuracy on ImageNet, spanning learning paradigms, training dataset sizes, and architectures. We also include CLIP, a model trained with caption supervision on over 400M text-pair images that has show impressive performance on several OoD benchmarks Radford et al. (2021). We also evaluate the zero-shot performance of CLIP trained on 2B images from the LAION dataset Ilharco et al. (2021).

We select SoTA supervised models of varying architectures. For ResNet-50, a CNN-based model, we use a ImageNet-1k pre-trained model based on the an improved training recipe from Wightman et al. (2021) achieving 80.4% top-1 accuracy on ImageNet and an ImageNet-21k weights from Ridnik et al. (2021) achieving 82.0% top-1 accuracy on ImageNet. For Vision Transformer (ViT), an attention-based model, we use an ImageNet-21k pre-trained ViT-B/16 achieving 83.97% and ImageNet-1k pretrained weights from Ridnik et al. (2021). For MLPMixer, a multi-layer percetron-based model, we use ImageNet-21k pretrained weights from Ridnik et al. (2021) and ImageNet-1k weights from Wightman (2019) using Base-16 architecture.

We also select several SoTA self-supervised learning models. For SimCLR (Chen et al., 2020b), a contrastive learning method, we select a ResNet-50 (CNN-based) backbone, trained on ImageNet-1k based on weights from Falcon & Cho (2020). For MAE He et al. (2022), a method based on a reconstruction objective, we select an attention-based ViT encoder. We use pre-trained weights from the official repo of He et al. (2022). For iBot Zhou et al. (2021), also a ViT-based model, we use ImageNet-1k and ImageNet-21k pre-trained weights from the official repo of Zhou et al. (2021).

All of the models are pre-trained using the standard data-augmentation pipeline proposed for each method. We load pretrained backbones for each model without modifying the pretraining data augmentations.

## 4 SoTA vision models are not robust to natural variations

We evaluate pretrained model's ability to generalize natural variation using two common protocols: linear evaluation and finetuning. We measure models' generalization by each model's classification accuracy for "canonical" settings and the same objects varying by one or more of the natural factors. Specifically, we define the generalization gap for each factor to be the (accuracy on held-out samples with randomly varying factor) - (accuracy on held-out sample in their canonical setting). Note that the canonical value of each factor is chosen arbitrarily, but fixed across all experiments such that the canonical value is simply the value which is dominant in the training data. We then evaluate these models on held-out objects which have factor values not seen in training, varying the values of one factor at a time. To control for differences in the performance of models on canonical data, we report the gap between the model's accuracy on the canonical and varying held-out sets.

**SoTA models are not robust to changes in pose, background, and size** While models reached strong performance on canonical data, Table 1 demonstrates that even SoTA models suffer considerable drops in performance when objects vary across factors. Models were particularly sensitive to changes in pose, background, and size, while models were largely robust to changes in position and lighting color. We hypothesize that this difference in robustness across factors may be related to how easily variation across a factor can be approximated by pixel-level augmentations. Both lighting color and position can be well approximated by color shift and translation, respectively.

(a) Linear evaluation gaps

| | Train accuracy | Held-out accuracy | Pose gap | Background gap | Size gap | Position gap | Lighting color gap | Average gap |
|---|---|---|---|---|---|---|---|---|
| CLIP | 80.65 | 72.22 | -42.40 | -25.43 | -19.84 | -5.70 | -2.65 | -19.20 |
| MAE | 30.12 | 21.11 | -13.77 | -10.77 | -6.79 | -2.30 | -2.36 | -7.20 |
| MLPMixer1k | 85.55 | 71.56 | -44.19 | -35.11 | -26.31 | -10.59 | -4.92 | -24.22 |
| MLPMixer21k | 91.59 | 80.37 | -43.25 | -26.83 | -20.32 | -5.45 | -1.53 | -19.48 |
| ResNet50-1k | 86.76 | 77.41 | -42.30 | -27.78 | -25.28 | -5.10 | -3.33 | -20.76 |
| ResNet50-21k | 92.89 | 76.22 | -35.49 | -23.20 | -14.85 | -0.51 | -1.04 | -15.02 |
| SimCLR | 91.69 | 73.33 | -51.05 | -33.93 | -28.01 | -5.85 | 1.11 | -23.55 |
| ViT-1k | 93.47 | 79.63 | -44.48 | -24.43 | -25.05 | -6.53 | -2.56 | -20.61 |
| ViT-21k | 91.82 | 78.89 | -39.87 | -20.38 | -26.73 | -6.97 | -0.89 | -18.97 |
| iBot-1k | 93.75 | 81.11 | -52.63 | -28.38 | -25.96 | -7.54 | -3.62 | -23.63 |
| iBot-21k | 93.60 | 82.96 | -52.90 | -31.66 | -30.46 | -7.14 | -1.06 | -24.64 |
| Average | 84.72 | 72.26 | -42.03 | -26.17 | -22.69 | -5.79 | -2.08 | -19.75 |

(b) Finetuning gaps

| | Train accuracy | Held-out accuracy | Pose gap | Background gap | Size gap | Position gap | Lighting color gap | Average gap |
|---|---|---|---|---|---|---|---|---|
| CLIP | 94.36 | 81.85 | -50.84 | -16.67 | -16.63 | -5.32 | -1.96 | -18.28 |
| MAE | 92.91 | 73.33 | -50.50 | -44.73 | -24.74 | -17.71 | -14.62 | -30.46 |
| MLPMixer1k | 90.73 | 80.37 | -51.10 | -25.76 | -23.13 | -7.08 | -3.17 | -22.05 |
| MLPMixer21k | 96.21 | 84.44 | -46.44 | -15.67 | -16.53 | -5.06 | -2.48 | -17.24 |
| ResNet50-1k | 95.61 | 80.96 | -50.63 | -18.41 | -20.64 | -6.66 | -4.06 | -20.08 |
| ResNet50-21k | 95.76 | 86.67 | -46.68 | -29.87 | -19.54 | -3.74 | -2.35 | -20.44 |
| SimCLR | 95.34 | 82.96 | -56.38 | -30.11 | -24.36 | -8.30 | -0.72 | -23.98 |
| ViT-1k | 96.17 | 84.44 | -46.61 | -14.36 | -16.98 | -3.34 | -1.66 | -16.59 |
| ViT-21k | 96.01 | 84.44 | -46.95 | -9.83 | -18.01 | -2.78 | -0.19 | -15.55 |
| iBot-1k | 94.56 | 84.81 | -53.22 | -25.57 | -23.99 | -5.82 | -3.01 | -22.32 |
| iBot-21k | 95.70 | 85.56 | -50.55 | -12.02 | -17.64 | -4.1 | -1.18 | -17.10 |
| Average | 94.85 | 82.71 | -49.99 | -22.09 | -20.20 | -6.36 | -3.22 | -20.37 |

Table 1: **SoTA models are not robust to common factors**: we show the drop in accuracy (negative values) relative to each model's held-out accuracy when an object is presented in its canonical setting for linear eval (a) and finetuning (b). All models are trained (linear eval or finetuning) on images of instances in their canonical setting.

In contrast, pose, background and size (relative to a fixed background) all require 3D manipulation of the object itself, and are therefore very difficult to replicate with pixel-level augmentations. We refer the interested reader to Tables A2, A3, A4 and A5 where we report the absolute as well as relative top-1 accuracy numbers associated with the gaps discussed here.

While finetuning consistently improved performance on canonical data (finetuned held-out canonical accuracy of 82.71% vs. 72.26% for linear), it actually hurt robustness relative to linear evaluation. Performance gaps on varying held-out instances *increased* after finetuning, demonstrating that while finetuning can improve in-distribution performance, it does so at the cost of generalization (Table 1).

## 4.1 Do architectural inductive biases matter?

**Learning objective is more impactful than architecture for robustness** In general, we found that robustness was similar across models with the notable exception of MAE. As shown in Table 1 (b), the MAE model is especially susceptible to changes in background, with a -44.7% drop compared to an average -19.4% for other models. MAE is also substantially more sensitive to position and lighting color. This sensitivity was not observed in other ViT based models, suggesting that it stems from differences in the training objective rather than the architecture. While all other models

use either supervised or InfoNCE based objectives, MAE uses a reconstruction objective. This focus on reconstruction may cause the model to pay closer attention to factors like background, position, and lighting color, as it is likely necessary to learn these correlations to effectively reconstruct.

Interestingly, the consistency across architectures also largely held for comparisons between CNNs and ViT based models, even for factors such as position (translation) for which CNNs are widely believed to be robust, although several recent works have suggested otherwise (Kayhan & Gemert, 2020; Liu et al., 2018; Bouchacourt et al., 2021; Biscione & Bowers, 2021; Ruderman et al., 2018; Zhang, 2019). We also found comparable gaps when evaluating CLIP using zero-shot classification, including CLIP trained on 2B LAION images (see Appendix G).

### 4.2 Are self-supervised models more robust?

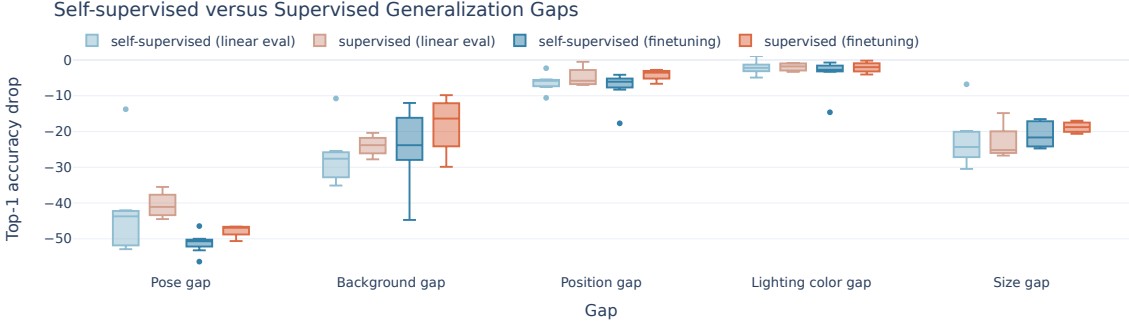

Figure 2: **Supervised models benefit more from finetuning than self-supervised models:** we compare generalization gaps for self-supervised and supervised models using box-plots. Models are trained only canonical instances.

Several recent works have suggested that pre-training with self-supervision may lead to increased robustness (Hendrycks et al., 2019; Geirhos et al., 2020). To test this, we compared the robustness of self-supervised models to supervised models in Figure 2. For linear evaluation, supervised models slightly outperformed SSL models on average, though SSL models were able to achieve a higher ceiling. In the finetuning setting, however, this difference is far more striking, suggesting that the robustness of supervised models benefits far more from finetuning than SSL models. Previous works (Fan et al., 2021; Chen et al., 2020a) noted that regular finetuning of the full network weights does not preserve the robustness self-supervised might have learned during unsupervised pretraining (e.g. with adversarial pretraining). This finding is in line with Kumar et al. (2022) which also studies in depth this phenomenon, both theoretically and experimentally, and find that finetuning leads to worse OOD performance compared to linear probing.

### 4.3 Can more training data improve robustness?

Recent works have shown that increasing the dataset size leads to substantial gains, especially for SSL models (Zhai et al., 2022; Goyal et al., 2021; Hoffmann et al., 2022; Kaplan et al., 2020). However, the effect of additional data on robustness remains unclear. To test this, in Figure 3, we focus on the comparison between ImageNet-21k (14 million training samples) and ImageNet-1k (1.2

million training samples). We found that for both finetuning and linear evaluation, models trained on ImageNet-21k were substantially more robust than those trained on ImageNet-1k (Figure 3). Interestingly, this effect was more pronounced in the context of finetuning than linear evaluation, with pose, size, and position benefitting most. Finetuning also led to less variance in accuracy drops across models, suggesting models robustness converges with finetuning compared to linear evaluation.

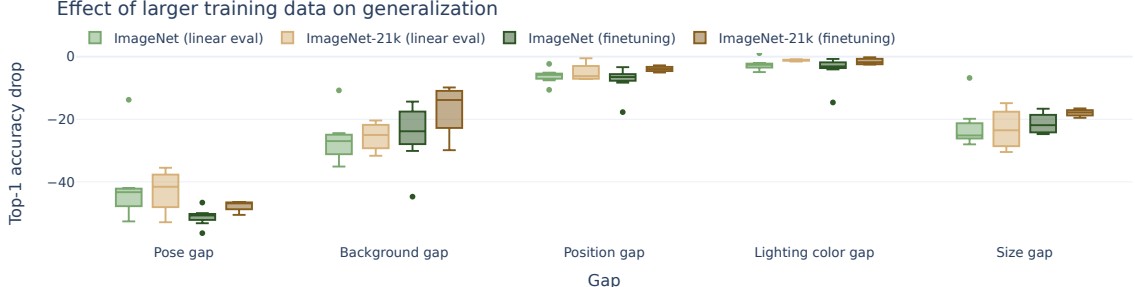

Figure 3: **Models trained on ImageNet-21k are more robust** compared to those trained on ImageNet-1k. We compare the effect training size for linear evaluation and finetuning on canonical samples.

## 5    Can models generalize variation from seeing variability in the training data?

In the previous section, we demonstrated that SoTA vision models struggle to generalize across several common factors such as pose or size. We also observed that pre-training on larger datasets (ImageNet-21k vs. ImageNet-1k) led to improved robustness, consistent with other results demonstrating the impact of additional data (Radford et al., 2021; Kaplan et al., 2020; Hoffmann et al., 2022; Zhai et al., 2022). Here, we study the extent to which models can generalize variability from training to held-out samples across three settings: 1) when all samples vary 2) when only some instances vary 3) when only some classes vary.

### 5.1    How much training variability is needed to close the generalization gaps?

We first measure the extent to which seeing all instances varying during training can close the generalization gaps. In order to introduce variation, we ensure a particular fraction of samples per instance feature diverse values (i.e., departing from canonical). We increase the amount of variability from 5 to 95% and evaluate how robustness to variability on novel instances at test time changes relative to the robustness of models trained only on data with canonical values for factors. During training, each sample has the given percent chance of the factor randomly varying. For example, 50% chance of varying pose means half of instances seen during training have a random pose while the other half are in the canonical pose. Figure 4 reports the effect of increasing variability on the generalization gaps for each factor. While all factors benefit from introducing variability, some factors such as pose and size still incur quite a large gap even with 50% variability. This result demonstrates that while incorporating variability during training improves robustness, the magnitude of this effect varies substantially across factors.

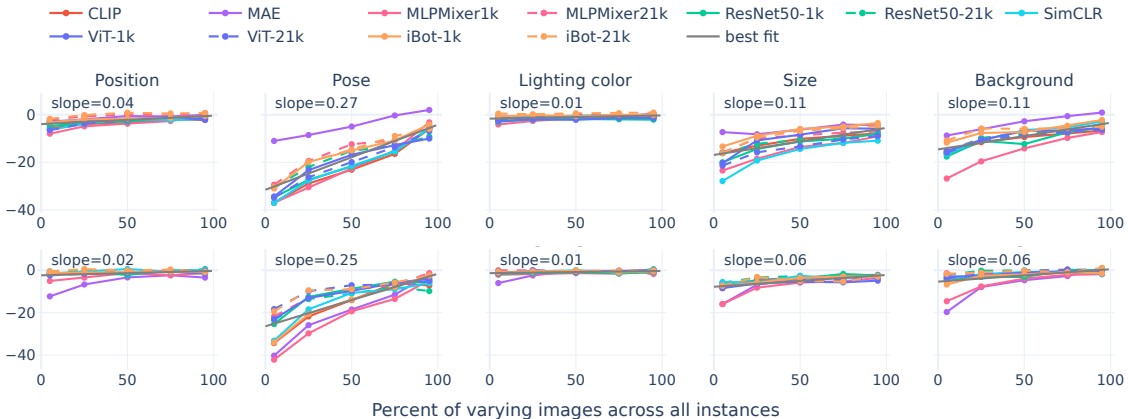

Figure 4: **Models require significant variation across all samples to close generalization gaps.** We show the generalization gaps as the variability across all samples increases using linear evaluation (top) and finetuning (bottom). The x-axis indicates how likely each training sample is to randomly vary for each factor. For example, 50% indicates half of samples seen during training randomly vary for the factor while the other half are in the canonical setting.

## 5.2 Can models generalize variation across instances?

The previous experiment measured whether introducing variability across all training instances helped robustness, but it remains unclear whether models can generalize variability in one set of instances to a different set of instances. This is analogous to the experiments of Alcorn et al. (2019) but extends their work to different levels of variability and additional factors. We thus introduce variability only for a subset of instances for each factor of variation. By contrast to Section 5.1, variability in % now refers to *the percentage of the instances that are seen undergoing variations*, while the rest of the instances are seen only with their canonical factors values during training. For example, 5% variability in pose means 5% of instances (objects) are seen in a random pose during training; the remaining 95% of instances (objects) are seen only in their canonical pose. This is a substantially more difficult generalization problem, as exemplified by the larger gaps observed in this setting. For example, as humans we don't need to see every dog rolling around to know dogs roll around. However, we found models struggle to generalize when the percent instances varying is low (<25%). Robustness improves with additional varying instances, with some factors reaching minimal gaps with as few as 50% of the training instances are seen varying (Figure A16). The pattern across factors was largely consistent, though both position and lighting color reached minimal gaps with comparatively less variability in training data, consistent with our previous observation that models are more robust to variance along these factors.

Interestingly, varying only a portion of *instances* led to substantial overfitting, especially when the proportion of varying instances is smaller than 50%. Compared to the original gap with no diversity in Section 4, the gaps are higher when initially introducing variability, and only return to their baseline values once sufficient variability is reached. For example, while position and lighting have gaps of -5% and -2% respectively with no variability (Table 1), their gaps when 5% of instances

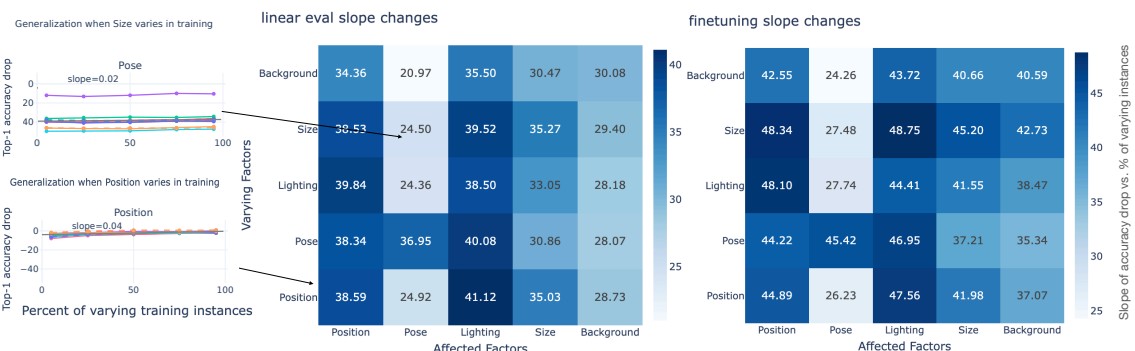

Figure 5: **Varying one factor can improve robustness to other factors**. We illustrate the cross-factor changes when a factor varies by plotting the change in gaps using the line of best fit as the number of varying instances increases.

vary are nearly -40% (Figure A16). This suggests that when the specific subset of training instances seen varying during learning is small, the model overfits to this small set of varying instances. We compare the effect finetuning vs. linear evaluation in Appendix Section I.

**Does training with instances varying for a single given factor improve robustness to variation in other factors?** Does robustness to a single factor provide broader robustness to other factors as well? To test this, we trained models with increasing amounts of variability for a single factor. In Figure 1 (last column), we show the average change in gap for factors other than factor varying during training. We find average effects of 1-3%. We further isolate this effect for each factor in the heatmaps show in Figure 5 by plotting the slope of the line of best fit across models as we increase the portion of varying instances seen during training. The diagonal of this heatmap represents cases where robustness evaluated for the same factor seen varying during training; off-diagonal entires measure changes for other factors. Inducing robustness to one factor consistently improved robustness for other factors by as much as 41% for linear evaluation and 48% for finetuning, though results varied across factor pairs. For example cross factor effects for pose are minor relative to those for position and lighting color. In fact, we found position and lighting color most helped each other, suggesting that the impact of position and lighting color variability are somewhat entangled.

**Does larger pretraining data improve generalization to varying held-out instances?** To test the importance of pretraining data size, we compared models trained on ImageNet-1k to those trained on ImageNet-21k. As can be seen in figure A11, ImageNet-21k pretraining consistently improves robustness compared to ImageNet-1k pretraining, whether for finetuning or linear evaluation.

### 5.3 Does training with instances varying in all factors improve robustness?

Training with variability for a single factor improves robustness both to the factor seen during training as well as other factors, but how does training with variability for all factors impact

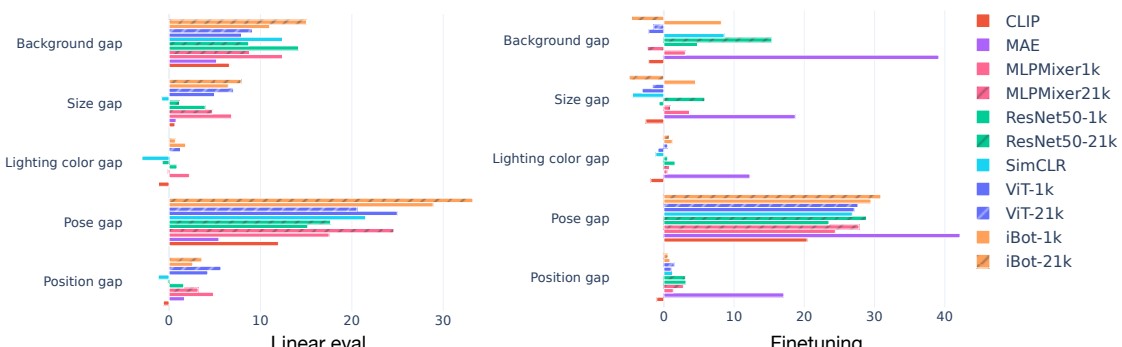

Figure 6: **Varying all factors during training improves robustness** We show show relative generalization gaps when all factors vary during training relative to no instances seeing varying (no variability). During finetuning or linear evaluation, half of training samples randomly vary across all factors; while the other half of samples are in the canoncial setting.

robustness? To test this, we selected random bases in the five-dimensional factor space and sampled images with random values along these bases during training. During finetuning or linear evaluation, half of training samples randomly vary across all factors; while the other half of samples are in the canonical setting. We report the change in the accuracy gap induced by incorporating factor variation during training (e.g., gap with no varying training factors - gap with all varying training factors) on held-out samples. Positive values indicate an improvement in robustness, while negative values indicate a decrease. We found that training with variability across all factors led to substantially improved robustness for most factors, though lighting color received no benefit, perhaps because its baseline robustness was already quite high (Figure 6). Interestingly, pose benefited the most from training with variability across all factors, despite being helped the least from the individual cross-factor variability, suggesting that while variability in other factors can improve pose robustness, variability across multiple factors simultaneously is necessary to induce noticeable improvements.

### 5.4 Can models generalize variation across classes?

We have shown that introducing variability during training improves robustness for new instances, and in some cases, for entirely different factors of variation. However, in all prior experiments, the class distribution was held constant such that models were only asked to generalize to new instances from the same class. Can models generalize robustness across classes? To test this, we trained models with variability only present for a single factor for half of the classes (randomly selected). For classes trained with variability, half of the instances within that class were seen varying for the given factor. Results are summarized in Tables A18 and Table 2.

**Models are significantly less robust when variation is only seen for some classes** We found significant gaps in generalization when only half of classes were seen with variability for each of the factor, as shown in Table 2. The average gap across all factors is -50% more than double the gaps observed where no variability is seen during training at all. This implies that when variation

|              | Position gap | Pose gap | Lighting color gap | Size gap | Background gap | Average gap |
|--------------|--------------|----------|--------------------|----------|----------------|-------------|
| CLIP         | -49.34       | -62.91   | -53.25             | -50.30   | -53.87         | -53.93      |
| MAE          | -37.17       | -48.30   | -51.14             | -42.35   | -49.21         | -45.64      |
| MLPMixer1k   | -47.33       | -60.10   | -51.26             | -46.41   | -52.36         | -51.49      |
| MLPMixer21k  | -46.18       | -62.77   | -50.28             | -47.92   | -47.79         | -50.99      |
| ResNet50-1k  | -45.66       | -53.22   | -43.62             | -49.03   | -35.44         | -45.39      |
| ResNet50-21k | -43.85       | -54.09   | -47.54             | -47.12   | -43.85         | -47.29      |
| SimCLR       | -45.49       | -59.69   | -46.59             | -44.24   | -29.39         | -45.08      |
| ViT-1k       | -48.13       | -66.16   | -51.90             | -49.01   | -48.39         | -52.72      |
| ViT-21k      | -47.17       | -61.91   | -49.93             | -45.59   | -47.56         | -50.43      |
| iBot-1k      | -46.53       | -65.76   | -49.82             | -50.61   | -51.47         | -52.84      |
| iBot-21k     | -48.22       | -67.85   | -51.46             | -54.14   | -49.33         | -54.20      |
| Average      | -45.91       | -60.25   | -49.71             | -47.88   | -46.24         | -50.00      |

Table 2: **Models have significant gaps in generalization when only half of classes were seen varying.** Table shows generalization gap differences between classes (27 randomly selected) seen with diversity and those not when finetuning.

is only observed for some classes, models generalize even more poorly and extends Alcorn et al. (2019)'s results demonstrating lack of generalization across instances at the class level. Our finding suggests we should develop explicit mechanisms for improving model generalization across classes.

**Models generalize equally poorly across classes, unless classes are very similar or very dissimilar to those seen varying during training**  It is possible that robustness can only be generalized across classes when the classes exceed some threshold similarity. To test this, we evaluated the cross-class robustness as a function of the distance between classes. Class distance was computed using a pre-trained word-embedding similarities (Honnibal & Montani, 2017). While the most dissimilar classes were harmed more, the majority of classes exhibited a similar detrimental effect regardless of their similarity to the training classes that were varying (Appendix Figure A23).

## 6 Discussion

In order to develop robust, trustworthy models which do not fail when presented with distribution shift, we much characterize the generalization capabilities of our current best approaches. In this work, we provided an extensive study of the robustness of SoTA models to naturally occurring variations, extending on previous work in a number of ways. Our experiments show that models fail to generalize to variations of a set factors on held-out instances unless a reasonable amount of variability is seen during training. Surprisingly, we found that providing the model with training variability on a single factor can help generalize to other factors which were not varied during training. However, models struggle to transfer their knowledge of variations across classes: when only some classes are seen undergoing variability in training, only very similar classes (not seen varying at training) benefited at evaluation. Finally, we found that inductive biases such as architecture and training paradigm had minimal impact on models' converged robustness, in contrast to the pre-training data size and the method of downstream training. We hope that our work, by shedding further light on the blind spots of state-of-the-art models, can help practitioners develop robust models that can confidently and safely be deployed at large.

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

| Parameter | Canonical value | Minimum value | Maximum value |
|---|---|---|---|
| Object rotation X,Y,Z | 0 | $-\frac{\pi}{2}$ | $\frac{\pi}{2}$ |
| Translation X,Y,Z | 0 | $-0.4$ | $0.4$ |
| Object scale | 1 | 0.4 | 1 |
| Lighting hue | 0.1 | 0 | 1 |
| Background | 0 | 0 | 25 |

Table A1: Values of the factors of variation used for the generation of our datasets. Each value is sampled uniformly from the given interval. Object rotationa are generated as Tayt-Bryan angles using extrinsic rotations. Background indicates the index of the background image used.

## A  Appendix

## B  Dataset

### B.1  Generation

As discussed in section 3.1, we select 50 models in 54 synsets from 3d Warehouse (Trimble Inc). The synsets used are the same as ShapeNetCore (Chang et al., 2015), which gives diversity in the objects' classes and makes a link to popular datasets for 3d models.

To generate the scenes and the renderings we rely on Blender (Blender Online Community) and use BlenderProc (Denninger et al., 2019) to simplify the automation of the generation process. We use background images coming from Li et al. (2022; 2021b;a) in order to generate more complex scenes. Particularly, we consider five types of background (sky, water, city, home and grass) and 5 backgrounds for each type, giving us 25 in total. We also change 4 scalar factors of variations that are position, pose, object size and lighting color. We describe the ranges used for each factor of variation in Table A1. We illustrate the ranges in Figures A1 to A10 by generating renderings for various objects.

We study three different datasets that are generated as follows:

- **One Factor:** We vary each factor independently using using 101 different scalar values equally spaced for scalar factors, or 25 for the backgrounds. This gives us 1.1M images.

- **Pairs of factors:** We vary pairs of factors using 11 different scalar values that are equally spaced for each factor, and 10 backgrounds. This gives us 3.1M images.

- **All factors:** We compute the full grid of possible combinations using 11 different scalar values that are equally spaced for each factor, and 10 backgrounds. We then sample 1000 combinations that we use on all objects. This gives us 2.7M images.

For all scenarios, when a factor of variation is not changed, we use the canonical values. The generation of all images takes around 1500 hours on a single NVIDIA V100 GPU, but can be easily parallelized.

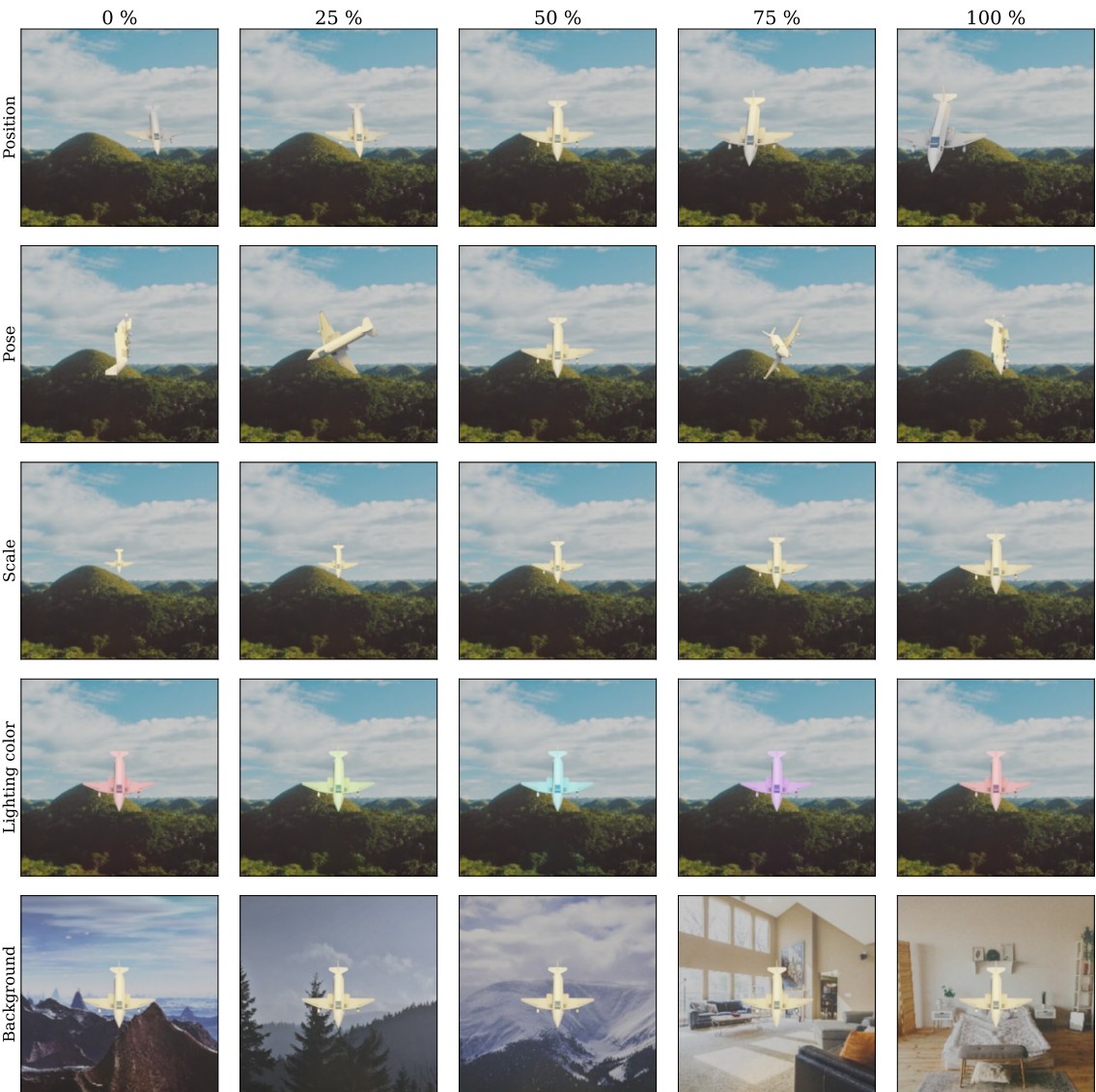

Figure A1: Examples from the dataset illustrating the different factors of variation.

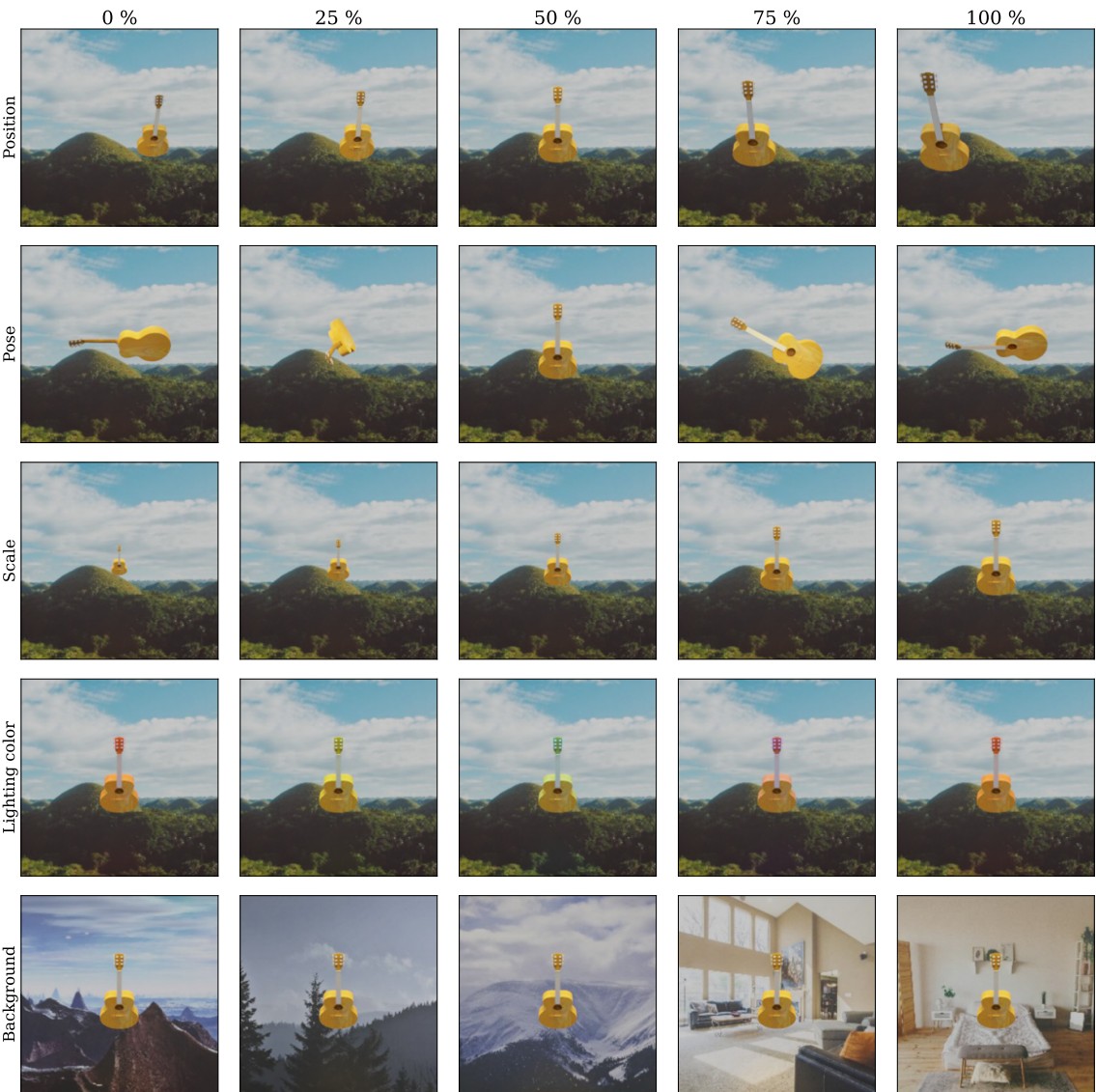

Figure A2: Examples from the dataset illustrating the different factors of variation.

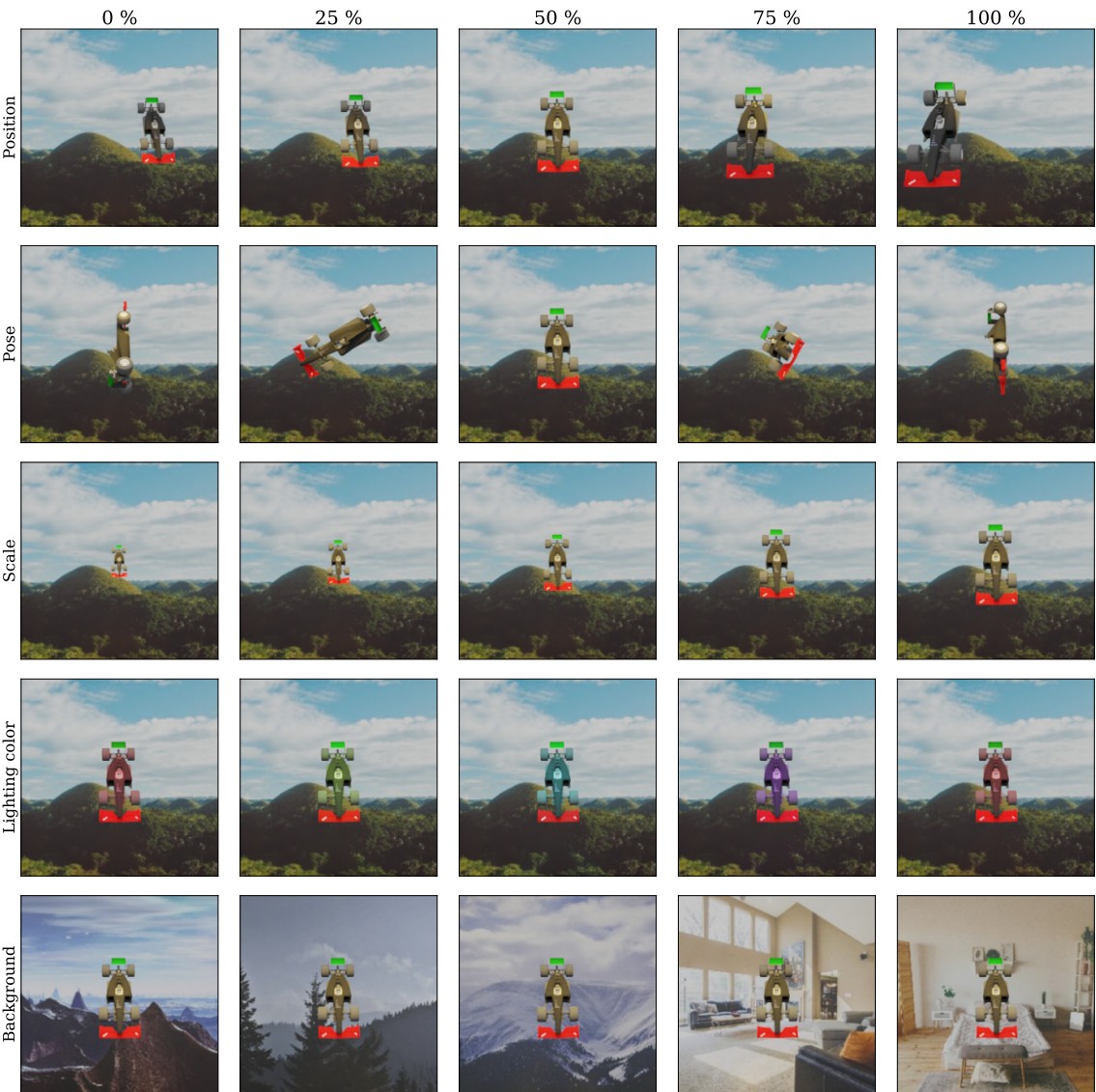

Figure A3: Examples from the dataset illustrating the different factors of variation.

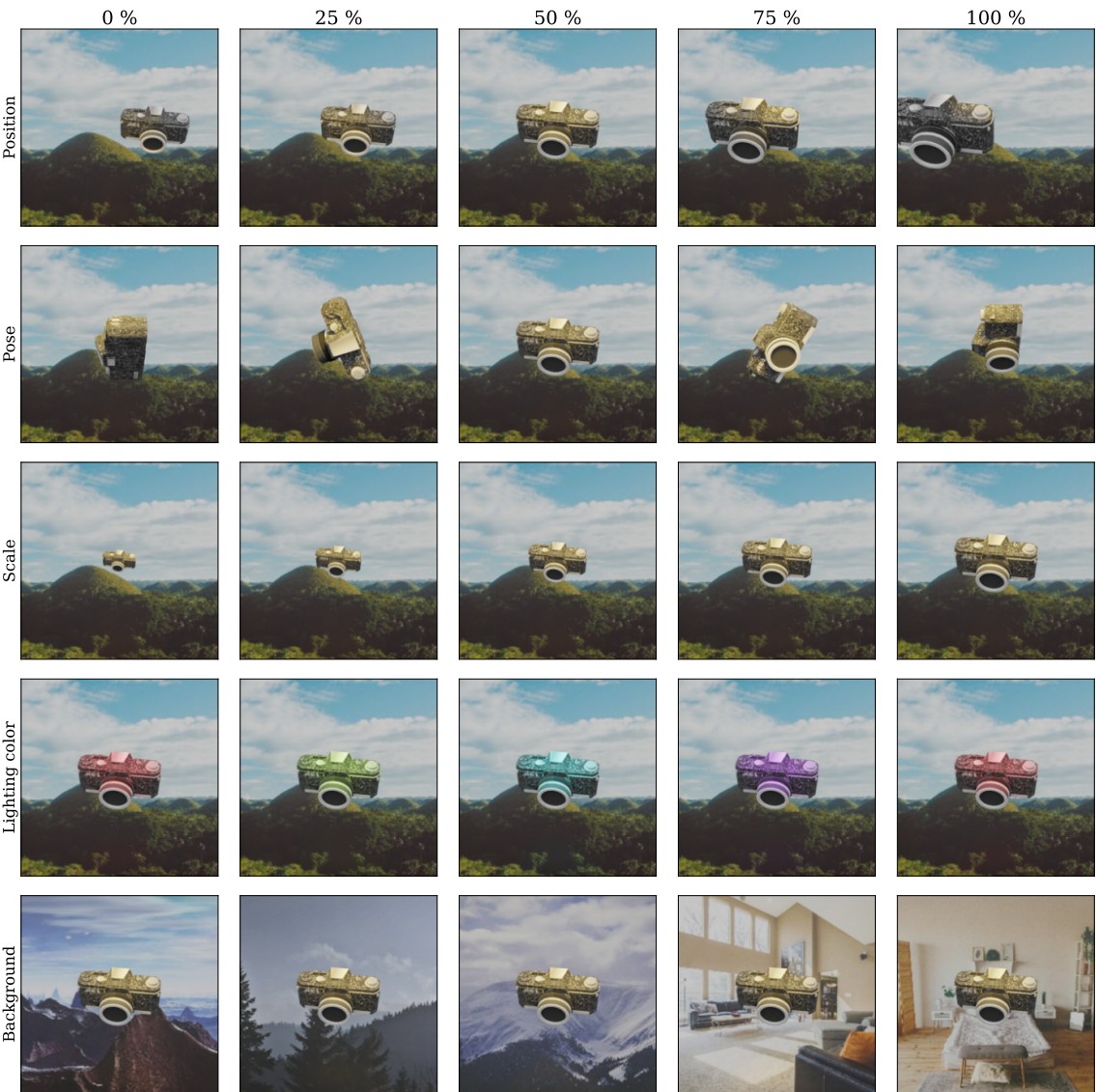

Figure A4: Examples from the dataset illustrating the different factors of variation.

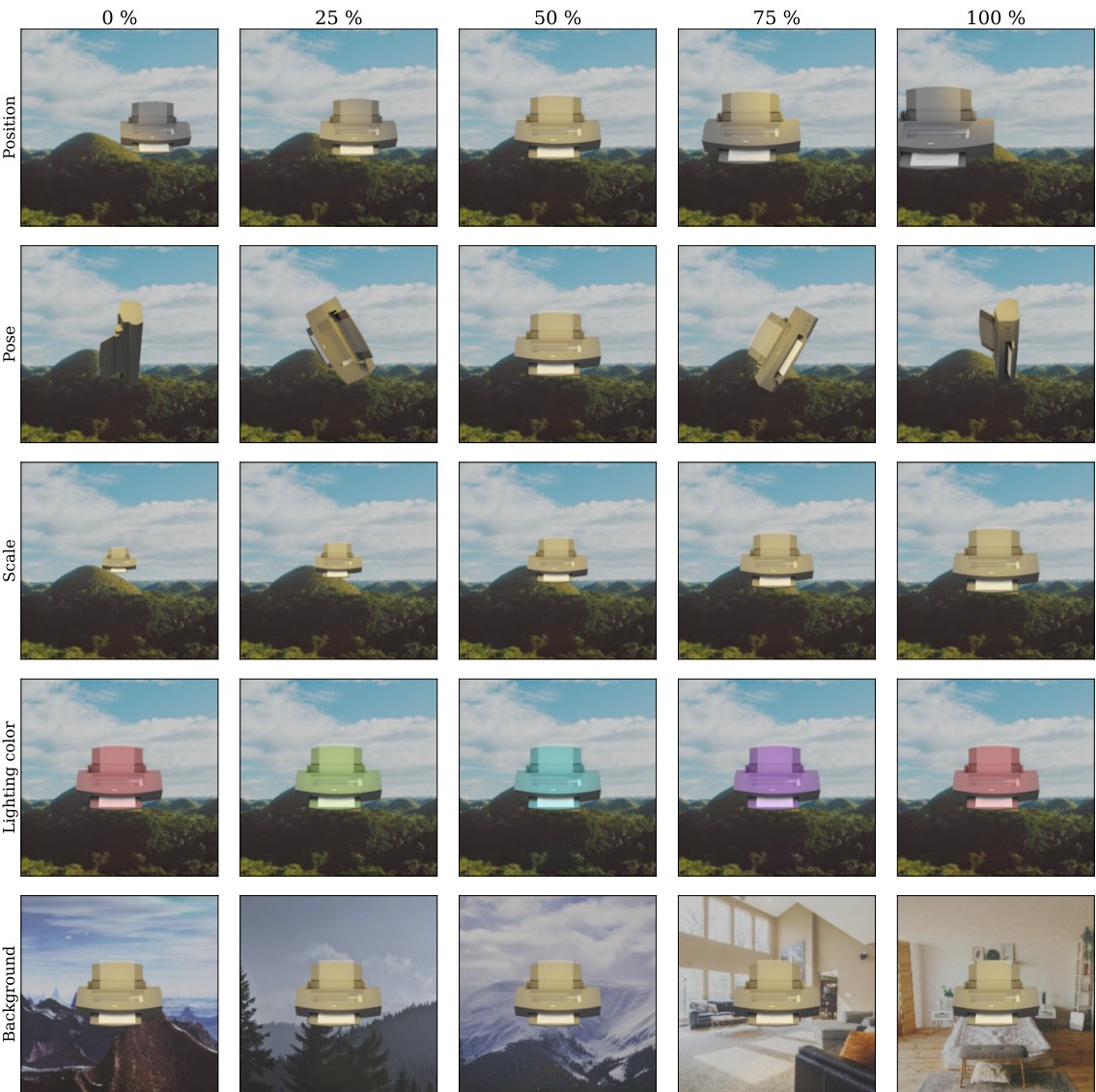

Figure A5: Examples from the dataset illustrating the different factors of variation.

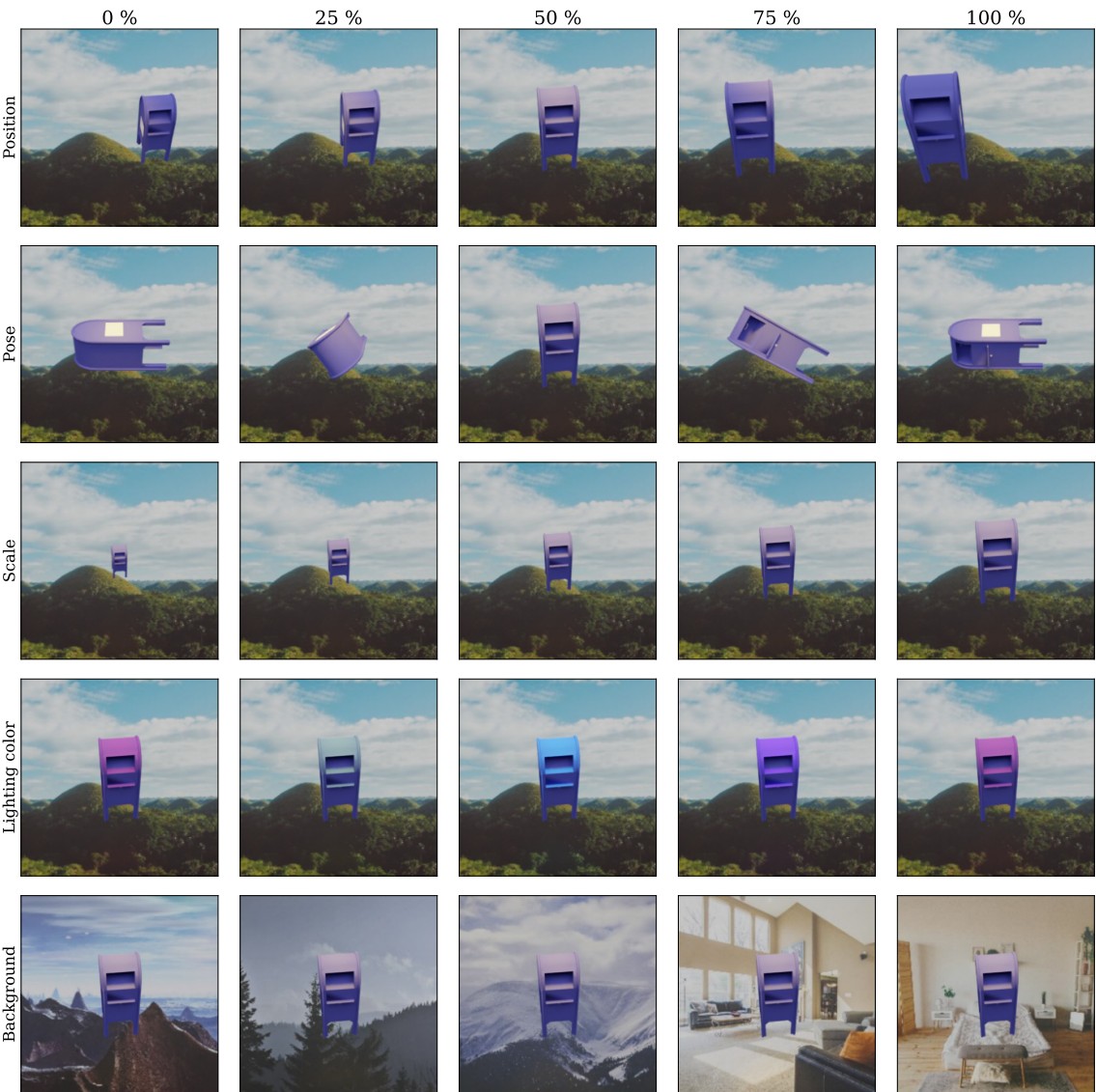

Figure A6: Examples from the dataset illustrating the different factors of variation.

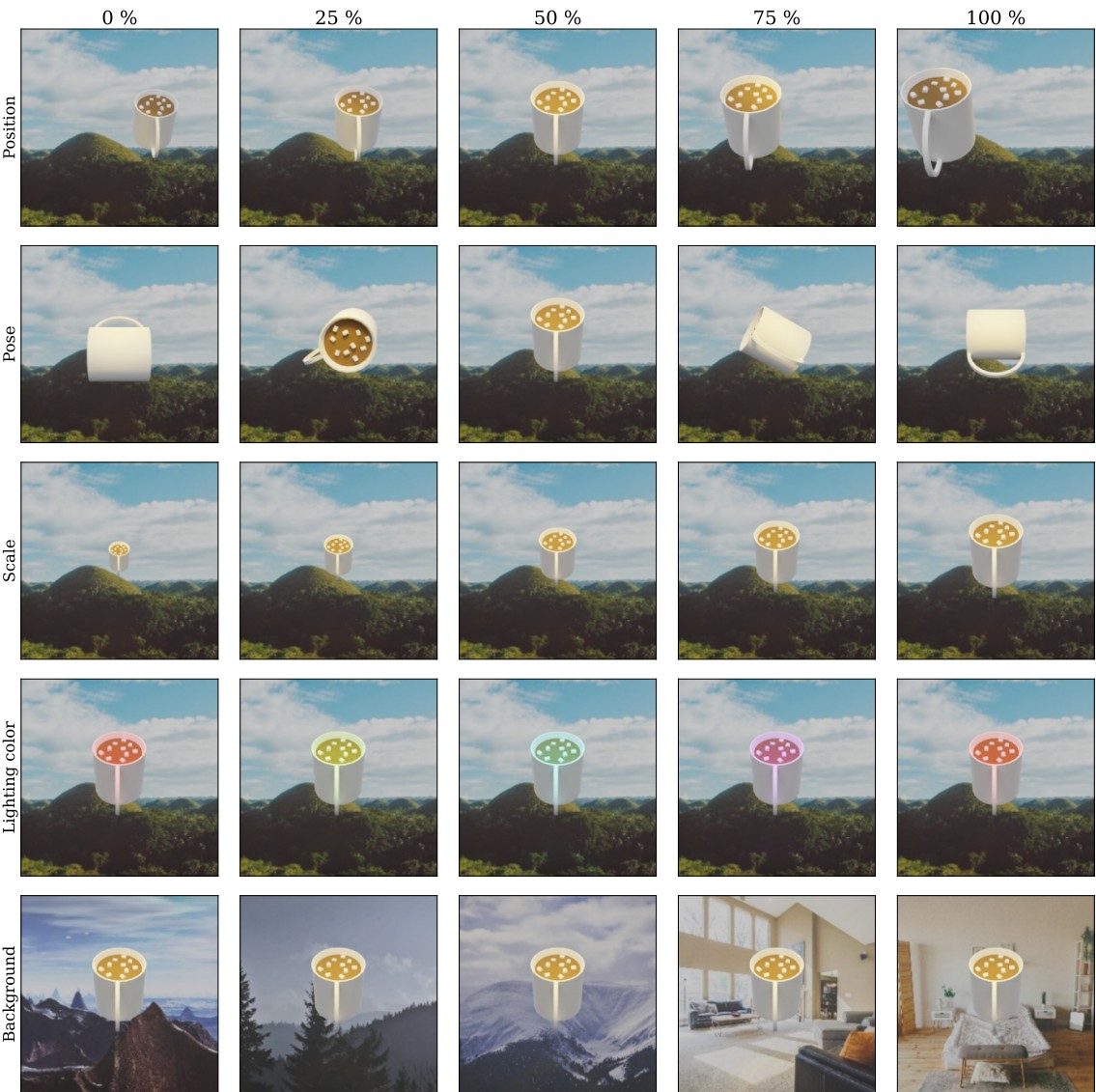

Figure A7: Examples from the dataset illustrating the different factors of variation.

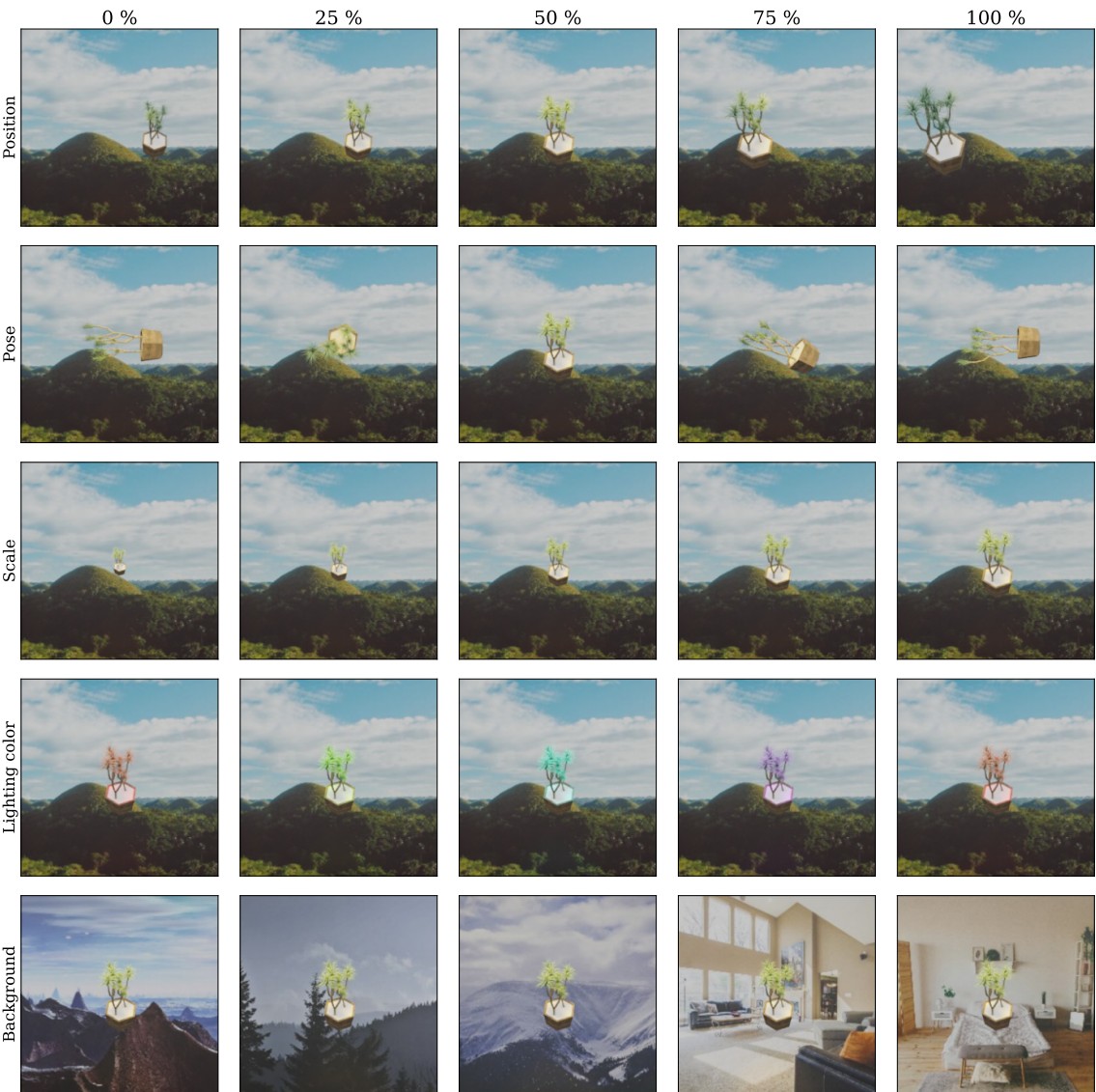

Figure A8: Examples from the dataset illustrating the different factors of variation.

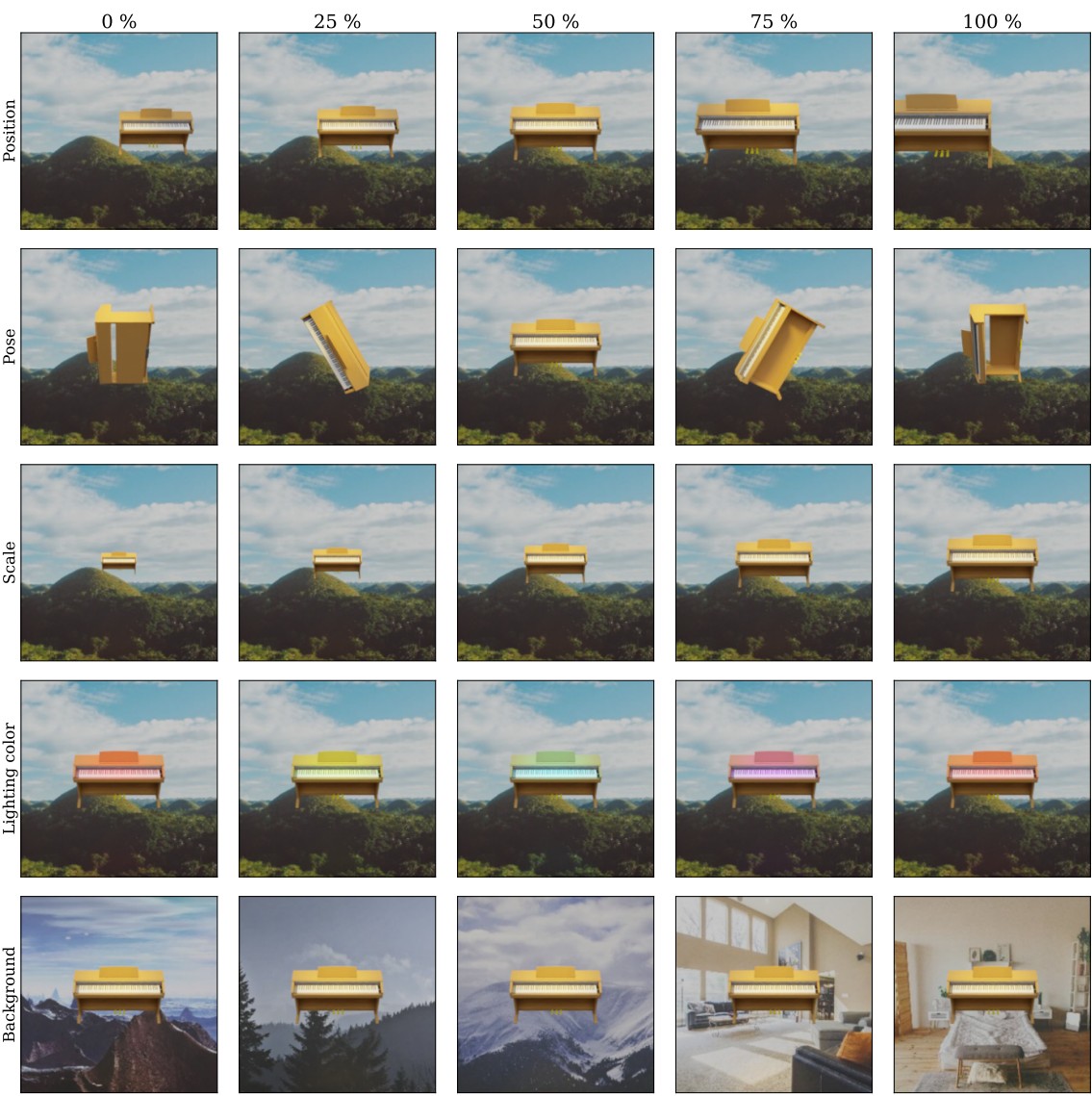

Figure A9: Examples from the dataset illustrating the different factors of variation.

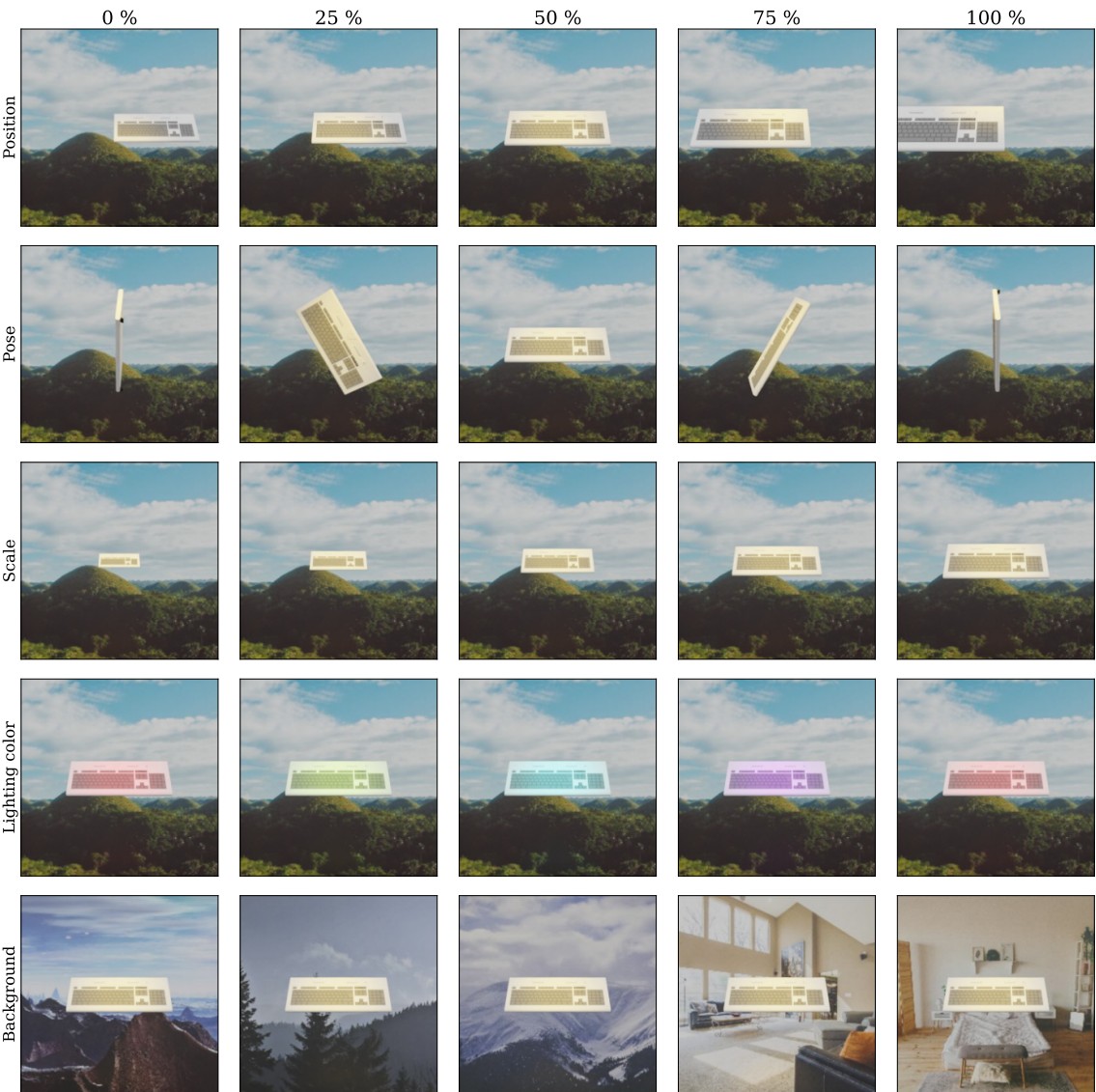

Figure A10: Examples from the dataset illustrating the different factors of variation.

| | Train canonical | Held-out canonical | Varying Pose | Varying Background | Varying Scale | Varying Position | Varying Lighting color |
|---|---|---|---|---|---|---|---|
| CLIP | 80.65 | 72.22 | 29.82 | 46.79 | 52.38 | 66.53 | 69.57 |
| MAE | 30.12 | 21.11 | 7.34 | 10.34 | 14.32 | 18.81 | 18.75 |
| MLPMixer1k | 85.55 | 71.56 | 27.37 | 36.45 | 45.24 | 60.97 | 66.63 |
| MLPMixer21k | 91.59 | 80.37 | 37.12 | 53.54 | 60.05 | 74.92 | 78.84 |
| ResNet50-1k | 86.76 | 77.41 | 35.11 | 49.63 | 52.13 | 72.31 | 74.08 |
| ResNet50-21k | 92.89 | 76.22 | 40.73 | 53.03 | 61.37 | 75.71 | 75.18 |
| SimCLR | 91.69 | 73.33 | 22.28 | 39.40 | 45.32 | 67.48 | 74.44 |
| ViT-1k | 93.47 | 79.63 | 35.15 | 55.20 | 54.58 | 73.10 | 77.07 |
| ViT-21k | 91.82 | 78.89 | 39.02 | 58.51 | 52.16 | 71.92 | 78.00 |
| iBot-1k | 93.75 | 81.11 | 28.48 | 52.73 | 55.15 | 73.57 | 77.49 |
| iBot-21k | 93.60 | 82.96 | 30.07 | 51.30 | 52.50 | 75.82 | 81.90 |

Table A2: Linear eval top-1 accuracy on canonical images and as each factor varies for held-out images.

| | Train canonical | Held-out canonical | Varying Pose | Varying Background | Varying Scale | Varying Position | Varying Lighting color |
|---|---|---|---|---|---|---|---|
| CLIP | 94.36 | 81.85 | 31.02 | 65.19 | 65.22 | 76.53 | 79.89 |
| MAE | 92.91 | 73.33 | 22.84 | 28.61 | 48.59 | 55.62 | 58.71 |
| MLPMixer1k | 90.73 | 80.37 | 29.27 | 54.61 | 57.25 | 73.29 | 77.20 |
| MLPMixer21k | 96.21 | 84.44 | 38.00 | 68.77 | 67.92 | 79.39 | 81.96 |
| ResNet50-1k | 95.61 | 80.96 | 30.34 | 62.55 | 60.32 | 74.31 | 76.90 |
| ResNet50-21k | 95.76 | 86.67 | 39.99 | 56.79 | 67.13 | 82.93 | 84.32 |
| SimCLR | 95.34 | 82.96 | 26.58 | 52.85 | 58.60 | 74.67 | 82.24 |
| ViT-1k | 96.17 | 84.44 | 37.83 | 70.09 | 67.46 | 81.10 | 82.79 |
| ViT-21k | 96.01 | 84.44 | 37.50 | 74.61 | 66.43 | 81.67 | 84.26 |
| iBot-1k | 94.56 | 84.81 | 31.60 | 59.25 | 60.82 | 79.00 | 81.80 |
| iBot-21k | 95.70 | 85.56 | 35.01 | 73.54 | 67.92 | 81.41 | 84.38 |

Table A3: Finetuning top-1 accuracy on canonical images and as each factor varies for held-out images.

## B.2 Dataset samples

## C Relative and absolute accuracy for each factor across models

In addition the performance gaps, we show in Tables A2 and A3 the absolute top-1 accuracy numbers associated with the gaps we report. We show accuracy for canonical instances (held-out and training) as well as performance as each factor varies (on held-out instances). These figures are for both linear evaluation and finetuned models.

We also show in Tables A4 and A5 the relative gap relative gap, that is, the gap value divided by canonical accuracy, for each model both in the linear evaluation and finetuning protocols.

| | Train canonical | Held-out canonical | Pose relative gap (%) | Background relative gap (%) | Size relative gap (%) | Position relative gap (%) | Lighting color relative gap (%) |
|---|---|---|---|---|---|---|---|
| CLIP | 80.65 | 72.22 | -58.71 | -35.21 | -27.47 | -7.89 | -3.67 |
| MAE | 30.12 | 21.11 | -65.23 | -51.00 | -32.18 | -10.91 | -11.16 |
| MLPMixer1k | 85.55 | 71.56 | -61.76 | -49.06 | -36.77 | -14.80 | -6.88 |
| MLPMixer21k | 91.59 | 80.37 | -53.81 | -33.39 | -25.28 | -6.78 | -1.91 |
| ResNet50-1k | 86.76 | 77.41 | -54.64 | -35.89 | -32.66 | -6.59 | -4.30 |
| ResNet50-21k | 92.89 | 76.22 | -46.56 | -30.43 | -19.48 | -0.67 | -1.37 |
| SimCLR | 91.69 | 73.33 | -69.62 | -46.27 | -38.20 | -7.98 | 1.52 |
| ViT-1k | 93.47 | 79.63 | -55.86 | -30.68 | -31.46 | -8.20 | -3.22 |
| ViT-21k | 91.82 | 78.89 | -50.54 | -25.84 | -33.88 | -8.83 | -1.12 |
| iBot-1k | 93.75 | 81.11 | -64.88 | -34.99 | -32.01 | -9.30 | -4.47 |
| iBot-21k | 93.60 | 82.96 | -63.76 | -38.16 | -36.71 | -8.61 | -1.28 |

Table A4: Linear eval top-1 accuracy on canonical images and the relative percent drop in accuracy as each factor varies for held-out images.

| | Train canonical | Held-out canonical | Pose relative gap (%) | Background relative gap (%) | Size relative gap (%) | Position relative gap (%) | Lighting color relative gap (%) |
|---|---|---|---|---|---|---|---|
| CLIP | 94.36 | 81.85 | -62.11 | -20.36 | -20.31 | -6.50 | -2.39 |
| MAE | 92.91 | 73.33 | -68.86 | -60.99 | -33.74 | -24.15 | -19.94 |
| MLPMixer1k | 90.73 | 80.37 | -63.58 | -32.05 | -28.77 | -8.81 | -3.95 |
| MLPMixer21k | 96.21 | 84.44 | -55.00 | -18.56 | -19.57 | -5.99 | -2.94 |
| ResNet50-1k | 95.61 | 80.96 | -62.53 | -22.74 | -25.50 | -8.22 | -5.02 |
| ResNet50-21k | 95.76 | 86.67 | -53.86 | -34.47 | -22.54 | -4.31 | -2.71 |
| SimCLR | 95.34 | 82.96 | -67.96 | -36.30 | -29.37 | -10.00 | -0.87 |
| ViT-1k | 96.17 | 84.44 | -55.20 | -17.00 | -20.11 | -3.96 | -1.96 |
| ViT-21k | 96.01 | 84.44 | -55.59 | -11.65 | -21.33 | -3.29 | -0.22 |
| iBot-1k | 94.56 | 84.81 | -62.74 | -30.15 | -28.29 | -6.86 | -3.55 |
| iBot-21k | 95.70 | 85.56 | -59.08 | -14.05 | -20.61 | -4.84 | -1.38 |

Table A5: Finetuning top-1 accuracy on canonical images and the relative percent drop in accuracy as each factor varies for held-out images.

## D    Aggregated performances for linear vs finetuning and 21k vs 1k

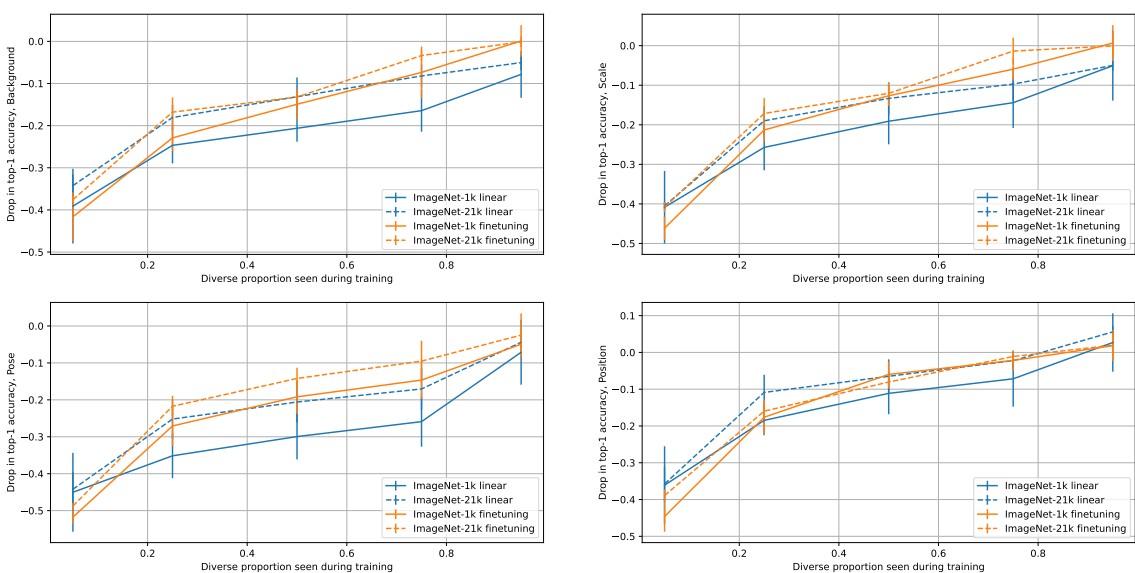

Figure A11: Drops in performance averaged over all methods when varying the proportion of varying examples seen during training.

As can be seen in figure A11, finetuning usually leads to lower drops in performance with high variability rates during training. However linear evaluation is more robust when diversity was not encountered during training. Pretraining on ImageNet21k always improves robustness compared to ImageNet-1k pretraining, whether in finetuning or in linear evaluation. It is worth noting that for translation robustness, all settings exhibit similar performance, and finetuning only benefits ImageNet-1k pretraining.

## E    Model sensitivity to variation strength

To further understand the sensitivity of model performance to the varying strengths of each variation, we measure the accuracy as we vary the variation strength. We perform the analysis for the best

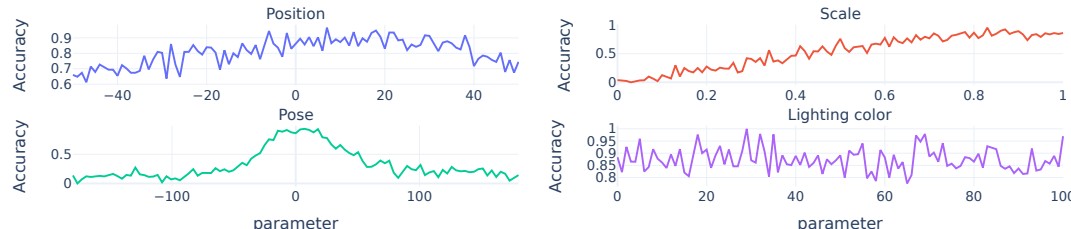

Figure A12: Sensitivity to increasing strength of variation for each factor for iBot pretrained on ImageNet-21k with linear evaluation.

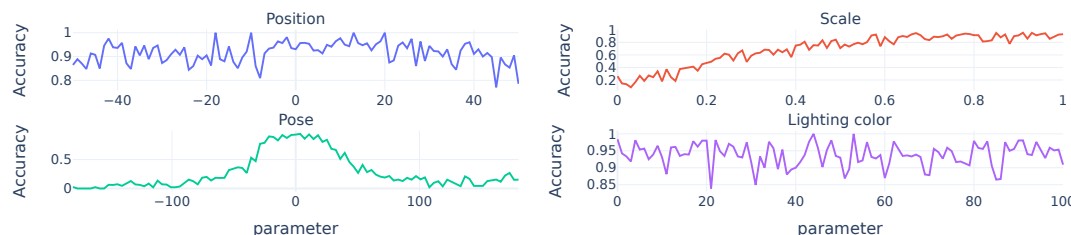

Figure A13: Sensitivity to increasing strength of variation for each factor for iBot pretrained on ImageNet-21k with finetuning.

performing model, iBot trained on ImageNet-21k, according to held-out accuracy. In Figure A12, we show the top-1 accuracy as the strength of each factor varies for iBot with linear evaluation. We find the canonical scale (1.0) and pose (0.0) have the highest accuracy with accuracy dropping as the variation strength increases. Position exhibits slightly higher accuracy with respect to the canonical (0.0), but we note the drop is more muted compared to that for pose and scale. Finally, in line with our earlier findings, since Lighting color gaps are quite small, we do not observe significant changes in accuracy as Lighting color strength varies. We show similar results in Figure A13 using the finetuning protocol.

As expected, Pose, Position and Scale factors show strong correlations between their strengths and the drop in performance. For Pose in particular, the accuracy is almost halved when going beyond 45 degree rotations. These findings are consistent with previous studies involving primates (Logothetis et al., 1994) who were asked to identify an object after rotation, among distractor objects. As we can see in Figure 5 A from Logothetis et al. (1994), primates exhibit an almost identical failure

| | Held-out canonical accuracy | Pose | Background | Scale | Position | Lighting color |
|---|---|---|---|---|---|---|
| BiT-JFT-300M | 73.09 | 32.71 | 47.29 | 54.11 | 69.26 | 72.81 |
| EfficientNet-L2-JFT-300M | 69.63 | 38.67 | 50.96 | 50.47 | 65.19 | 70.19 |

Table A6: Linear eval top-1 accuracy on canonical images and as each factor varies for held-out images.

| | Held-out canonical accuracy | Pose | Background | Scale | Position | Lighting color |
|---|---|---|---|---|---|---|
| BiT-JFT-300M | 79.01 | 32.54 | 58.94 | 62.55 | 74.40 | 78.09 |
| EfficientNet-L2-JFT-300M | 80.49 | 40.93 | 69.87 | 71.82 | 78.13 | 71.33 |

Table A7: Finetuning top-1 accuracy on canonical images and as each factor varies for held-out images.

mode as neural networks when varying the object's pose. Nonetheless the tasks are not identical and identifiability is arguably harder than classification due to its more fine-grained nature.

## F   Additional Models Trained on JFT-300M

In addition to the models discussed in the main paper, here we also evaluate two models trained on Google's properietary JFT datasets comprised of 300M images Sun et al. (2017). We performed both linear evaluation and finetuning for BiT, Big Transfer model, as well as EfficientNetL2 (ns 475) from retrained on JFT 300M Sun et al. (2017). We trained each model for 2k steps with a batch size of 32 for BiT and 8 for EfficientNetL2 to accommodate GPU memory. In Table A6 we show linear evaluation results and in Table A7 finetuning. Similar other pretrained models we examined, we observe comparable performance drops relative to each model's canonical performance. We note the higher held-out accuracy after finetuning as expected. This suggests even models trained on JFT exhibit similar performance drops as the set of factors we consider vary.

## G   CLIP Zero Shot Classification

We also evaluate CLIP's robustness using zero-shot classification. We assess both the standard Open AI CLIP model as well as CLIP trained on 2B LAION images. We prompt the model using "a photo of a []". CLIP with LAION-2B accuracies are 31.9% for canonical, 15.9% when pose varies, 18.2% when scale varies, 31.9% when lighting color, and 26.8% background varies. CLIP with trained on 400M images has canonical 30.1%, pose 16.0%, scale 18.4%, lighting color 28.3%, and background 23.6% accuracy.

We examine other prompts ("[], an inanimate object", "a photo of a [], an inanimate object", "[], a household item or vehicle") and observe similar classification performance (25-31% accuracy) using these variants.

## H   Cross Factor effects when varying all instances

The cross factor effects when all instances vary with increasing diversity levels are shown in Figures A14 and A15.

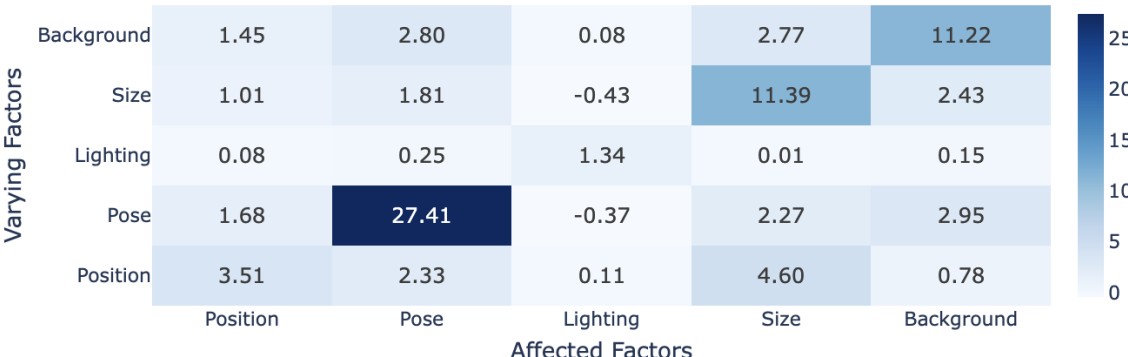

Figure A14: Cross factor changes when the given factor is varying for linear evaluation

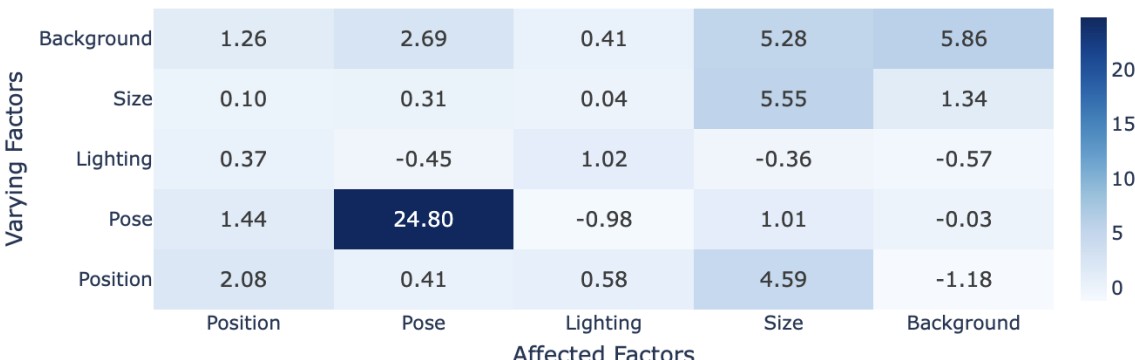

Figure A15: Cross factor changes when the given factor is varying for finetuning

# I  Varying a subset of instances during training

We show the effect of increasing the number of instances seen varying during training. In Figure A16 we show the effect of each factor. We break down the effect by factor in figures A17 for linear evaluation and finetuning A19. In addition, we show the overall accuracy in tables A8, A12, A9, A10. The val canonical column corresponds to held-out accuracy for canonical and the val diverse corresponds to the accuracy for a changing factor.

**Finetuning vs. linear evaluation as variability increases during training**  To summarize these results across factors, for each subplot, we compute a linear fit to the average model curve and compute its slope. Models with higher slopes are more sensitive to the fraction of instances seen varying during training, while lower slopes indicate models which have the same generalization gap regardless of how much instances was presented varying during training. The average slope across factors and models for finetuning was $0.359 \pm 0.035$ vs. $0.441 \pm 0.020$ for linear evaluation (mean $\pm$ std). This result demonstrates that, while both benefit from increasing the percentage of instances seeing varying during training, this effect is much more pronounced for finetuning, providing further evidence that the impact of supervision is larger for finetuned models, likely because of the increased expressivity introduced by allowing all the weights to change.

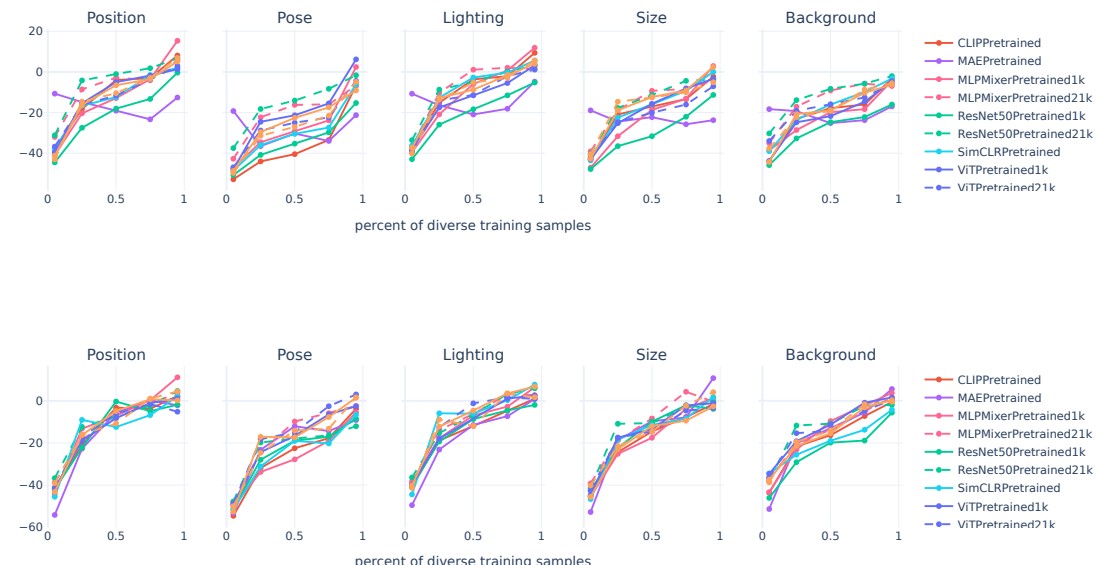

(a) Training with increasing percentage of variability across all instances using finetuning

Figure A16: Training with increasing percentage of instances seen varying during training using linear evaluation (top) and finetuning (bottom).

| train_prop_to_vary | train_canonical_top_1_accuracy | | | | | val_canonical_top_1_accuracy | | | | | val_diverse_Translation_top_1_accuracy | | | | |
|---|---|---|---|---|---|---|---|---|---|---|---|---|---|---|---|
| model | 0.05 | 0.25 | 0.50 | 0.75 | 0.95 | 0.05 | 0.25 | 0.50 | 0.75 | 0.95 | 0.05 | 0.25 | 0.50 | 0.75 | 0.95 |
| CLIPPretrained | 92.24% | 91.37% | 92.25% | 94.01% | 96.33% | 77.78% | 75.93% | 68.52% | 68.52% | 58.89% | 39.82% | 57.70% | 64.51% | 65.45% | 66.98% |
| MAEPretrained | 52.54% | 55.93% | 60.44% | 67.91% | 83.79% | 20.37% | 27.78% | 33.33% | 38.89% | 27.78% | 9.67% | 12.91% | 14.26% | 15.56% | 15.18% |
| MLPMixerPretrained1k | 94.66% | 93.98% | 94.58% | 95.75% | 97.25% | 77.78% | 74.07% | 72.22% | 66.67% | 48.15% | 36.86% | 53.70% | 59.35% | 63.05% | 63.46% |
| MLPMixerPretrained21k | 94.95% | 95.09% | 95.19% | 96.02% | 97.13% | 75.93% | 75.93% | 74.07% | 77.78% | 70.37% | 43.89% | 67.36% | 71.25% | 73.69% | 76.18% |
| ResNet50Pretrained1k | 95.00% | 94.58% | 94.83% | 95.79% | 97.23% | 88.89% | 90.74% | 87.04% | 83.33% | 70.37% | 44.35% | 63.26% | 68.99% | 70.04% | 70.01% |
| ResNet50Pretrained21k | 95.51% | 95.30% | 95.90% | 96.27% | 97.39% | 77.78% | 72.22% | 74.07% | 74.07% | 70.37% | 46.61% | 68.04% | 73.02% | 75.90% | 77.13% |
| SimCLRPretrained | 96.13% | 95.74% | 96.22% | 96.82% | 97.50% | 81.48% | 79.63% | 81.48% | 72.22% | 70.00% | 43.73% | 62.96% | 69.10% | 70.63% | 72.02% |
| ViTPretrained1k | 95.79% | 96.18% | 96.37% | 96.80% | 97.75% | 88.89% | 83.33% | 77.78% | 77.41% | 75.93% | 49.34% | 67.92% | 72.74% | 75.65% | 77.22% |
| ViTPretrained21k | 95.43% | 95.01% | 95.62% | 96.39% | 97.50% | 83.33% | 83.33% | 83.33% | 77.78% | 72.22% | 46.59% | 67.21% | 71.71% | 74.02% | 75.17% |
| iBotPretrained1k | 96.67% | 96.43% | 96.49% | 97.01% | 97.66% | 81.48% | 81.48% | 79.63% | 79.63% | 72.22% | 40.27% | 65.23% | 73.10% | 76.06% | 77.33% |
| iBotPretrained21k | 96.84% | 96.30% | 96.44% | 96.92% | 97.61% | 90.74% | 85.19% | 87.04% | 81.48% | 72.22% | 47.69% | 70.55% | 76.57% | 78.53% | 79.04% |

Table A8: Position varying linear eval top-1 accuracy across multiple percentages of varying training instances.

| train_prop_to_vary | train_canonical_top_1_accuracy | | | | | val_canonical_top_1_accuracy | | | | | val_diverse_Rotation_top_1_accuracy | | | | |
|---|---|---|---|---|---|---|---|---|---|---|---|---|---|---|---|
| model | 0.05 | 0.25 | 0.50 | 0.75 | 0.95 | 0.05 | 0.25 | 0.50 | 0.75 | 0.95 | 0.05 | 0.25 | 0.50 | 0.75 | 0.95 |
| CLIPPretrained | 91.70% | 91.43% | 92.42% | 93.98% | 96.28% | 81.48% | 85.19% | 85.19% | 79.63% | 55.56% | 28.64% | 41.16% | 44.72% | 46.17% | 48.85% |
| MAEPretrained | 53.90% | 57.24% | 61.28% | 68.44% | 83.90% | 25.93% | 44.44% | 38.89% | 42.59% | 29.63% | 6.68% | 7.93% | 8.58% | 8.63% | 8.36% |
| MLPMixerPretrained1k | 94.24% | 93.76% | 94.74% | 95.72% | 97.28% | 74.07% | 74.07% | 72.22% | 70.37% | 46.30% | 26.84% | 39.47% | 43.04% | 46.51% | 48.73% |
| MLPMixerPretrained21k | 94.49% | 94.81% | 95.03% | 95.88% | 97.13% | 79.63% | 77.78% | 77.78% | 79.63% | 72.22% | 36.90% | 55.46% | 61.57% | 63.67% | 65.50% |
| ResNet50Pretrained1k | 94.72% | 94.56% | 94.89% | 95.74% | 97.18% | 85.19% | 87.04% | 85.19% | 81.48% | 68.52% | 34.44% | 46.22% | 49.90% | 51.73% | 53.24% |
| ResNet50Pretrained21k | 95.51% | 94.96% | 95.69% | 96.12% | 97.30% | 77.78% | 75.93% | 75.93% | 72.22% | 66.67% | 40.29% | 57.68% | 61.83% | 63.91% | 65.04% |
| SimCLRPretrained | 95.74% | 95.63% | 96.16% | 96.80% | 97.51% | 81.48% | 83.33% | 83.33% | 83.33% | 62.96% | 32.94% | 47.35% | 52.88% | 55.97% | 56.91% |
| ViTPretrained1k | 95.71% | 95.76% | 96.09% | 96.79% | 97.67% | 87.04% | 79.63% | 81.48% | 79.63% | 59.26% | 39.13% | 55.09% | 60.14% | 64.14% | 65.42% |
| ViTPretrained21k | 95.23% | 94.89% | 95.68% | 96.42% | 97.45% | 85.19% | 83.33% | 83.33% | 83.33% | 66.67% | 38.25% | 54.50% | 58.34% | 60.94% | 62.28% |
| iBotPretrained1k | 96.39% | 96.22% | 96.41% | 96.96% | 97.56% | 83.33% | 81.48% | 81.48% | 79.63% | 72.22% | 34.62% | 51.88% | 58.88% | 62.19% | 63.11% |
| iBotPretrained21k | 96.44% | 96.09% | 96.42% | 96.92% | 97.61% | 90.74% | 88.89% | 90.74% | 87.04% | 72.22% | 41.03% | 57.48% | 63.69% | 65.49% | 67.34% |

Table A9: Pose linear eval top-1 accuracy across multiple percentages of varying training instances.

| train_prop_to_vary | train_canonical_top_1_accuracy | | | | | val_canonical_top_1_accuracy | | | | | val_diverse_Spot hue_top_1_accuracy | | | | |
|---|---|---|---|---|---|---|---|---|---|---|---|---|---|---|---|
| model | 0.05 | 0.25 | 0.50 | 0.75 | 0.95 | 0.05 | 0.25 | 0.50 | 0.75 | 0.95 | 0.05 | 0.25 | 0.50 | 0.75 | 0.95 |
| CLIPPretrained | 92.49% | 91.60% | 92.46% | 94.06% | 96.32% | 77.78% | 75.93% | 70.37% | 70.37% | 59.26% | 39.61% | 60.78% | 66.72% | 68.27% | 68.64% |
| MAEPretrained | 52.26% | 55.43% | 60.36% | 67.78% | 83.62% | 22.22% | 31.48% | 37.04% | 37.04% | 20.37% | 11.51% | 15.09% | 16.10% | 18.91% | 15.58% |
| MLPMixerPretrained1k | 95.17% | 94.20% | 94.98% | 95.86% | 97.30% | 79.63% | 77.78% | 70.37% | 66.67% | 57.04% | 40.09% | 56.90% | 64.43% | 67.35% | 68.93% |
| MLPMixerPretrained21k | 94.89% | 95.29% | 95.44% | 96.13% | 97.20% | 81.48% | 79.63% | 72.22% | 74.07% | 76.30% | 44.68% | 69.30% | 73.32% | 76.12% | 77.75% |
| ResNet50Pretrained1k | 95.26% | 94.81% | 95.03% | 95.80% | 97.23% | 87.04% | 88.89% | 87.04% | 83.33% | 77.78% | 44.08% | 62.96% | 68.67% | 71.81% | 72.54% |
| ResNet50Pretrained21k | 95.91% | 95.45% | 96.00% | 96.28% | 97.41% | 77.78% | 75.93% | 75.93% | 72.22% | 71.11% | 44.22% | 67.22% | 70.95% | 72.38% | 74.39% |
| SimCLRPretrained | 96.30% | 95.91% | 96.27% | 96.84% | 97.59% | 79.63% | 75.93% | 74.07% | 74.07% | 66.67% | 43.36% | 63.22% | 71.32% | 73.72% | 72.26% |
| ViTPretrained1k | 95.99% | 96.36% | 96.54% | 96.95% | 97.80% | 90.74% | 87.04% | 83.33% | 81.48% | 74.07% | 52.53% | 69.53% | 71.81% | 76.01% | 77.28% |
| ViTPretrained21k | 95.57% | 95.39% | 95.84% | 96.51% | 97.59% | 85.19% | 81.48% | 83.33% | 77.78% | 75.93% | 46.01% | 67.50% | 71.97% | 75.75% | 77.06% |
| iBotPretrained1k | 96.50% | 96.55% | 96.74% | 97.12% | 97.70% | 79.63% | 81.48% | 81.48% | 77.78% | 74.07% | 42.32% | 68.61% | 72.75% | 76.28% | 78.04% |
| iBotPretrained21k | 97.01% | 96.42% | 96.65% | 97.04% | 97.64% | 88.89% | 87.04% | 83.33% | 83.33% | 75.93% | 48.83% | 73.37% | 78.09% | 80.39% | 81.55% |

Table A10: Spot hue linear eval top-1 accuracy across multiple percentages of varying training instances.

| train_prop_to_vary | train_canonical_top_1_accuracy | | | | | val_canonical_top_1_accuracy | | | | | val_diverse_Scale_top_1_accuracy | | | | |
|---|---|---|---|---|---|---|---|---|---|---|---|---|---|---|---|
| model | 0.05 | 0.25 | 0.50 | 0.75 | 0.95 | 0.05 | 0.25 | 0.50 | 0.75 | 0.95 | 0.05 | 0.25 | 0.50 | 0.75 | 0.95 |
| CLIPPretrained | 91.81% | 91.51% | 92.26% | 93.90% | 96.25% | 79.63% | 72.22% | 74.07% | 70.37% | 61.11% | 36.80% | 51.29% | 57.05% | 56.99% | 58.80% |
| MAEPretrained | 51.98% | 56.15% | 60.87% | 68.07% | 84.12% | 25.93% | 35.19% | 33.33% | 37.04% | 35.19% | 7.01% | 10.74% | 11.05% | 11.29% | 11.44% |
| MLPMixerPretrained1k | 94.27% | 93.55% | 94.47% | 95.59% | 97.17% | 79.63% | 77.78% | 70.37% | 68.52% | 53.70% | 32.50% | 46.15% | 51.66% | 55.30% | 56.62% |
| MLPMixerPretrained21k | 94.86% | 94.98% | 95.08% | 95.93% | 97.09% | 79.63% | 79.63% | 75.93% | 75.93% | 74.07% | 40.55% | 60.87% | 66.63% | 67.52% | 70.03% |
| ResNet50Pretrained1k | 94.89% | 94.57% | 94.84% | 95.66% | 97.16% | 87.04% | 90.74% | 90.74% | 83.33% | 72.22% | 39.22% | 54.23% | 59.10% | 61.27% | 60.89% |
| ResNet50Pretrained21k | 95.51% | 95.28% | 95.77% | 96.08% | 97.35% | 81.48% | 77.78% | 77.78% | 74.07% | 74.07% | 41.11% | 60.06% | 66.41% | 69.69% | 70.33% |
| SimCLRPretrained | 95.91% | 95.53% | 96.09% | 96.66% | 97.48% | 83.33% | 77.41% | 77.78% | 72.22% | 62.96% | 39.79% | 54.93% | 61.81% | 62.93% | 62.96% |
| ViTPretrained1k | 95.65% | 96.01% | 96.18% | 96.69% | 97.69% | 87.04% | 83.33% | 79.63% | 75.93% | 72.22% | 44.10% | 58.13% | 63.91% | 67.85% | 68.82% |
| ViTPretrained21k | 95.06% | 94.87% | 95.47% | 96.30% | 97.38% | 83.33% | 83.33% | 83.33% | 81.48% | 72.22% | 41.40% | 58.53% | 63.25% | 65.43% | 65.14% |
| iBotPretrained1k | 96.58% | 96.37% | 96.48% | 96.98% | 97.60% | 84.07% | 77.78% | 79.63% | 77.78% | 66.67% | 41.34% | 58.93% | 67.24% | 68.66% | 69.08% |
| iBotPretrained21k | 96.92% | 96.18% | 96.39% | 96.86% | 97.53% | 85.19% | 77.78% | 81.48% | 81.48% | 75.93% | 44.59% | 63.11% | 68.90% | 71.45% | 70.82% |

Table A11: Scale linear eval top-1 accuracy across multiple percentages of varying training instances

| train_prop_to_vary | train_canonical_top_1_accuracy | | | | | val_canonical_top_1_accuracy | | | | | val_diverse_Background path_top_1_accuracy | | | | |
|---|---|---|---|---|---|---|---|---|---|---|---|---|---|---|---|
| model | 0.05 | 0.25 | 0.50 | 0.75 | 0.95 | 0.05 | 0.25 | 0.50 | 0.75 | 0.95 | 0.05 | 0.25 | 0.50 | 0.75 | 0.95 |
| CLIPPretrained | 90.37% | 91.88% | 92.73% | 94.36% | 96.70% | 85.19% | 77.78% | 77.78% | 77.78% | 66.67% | 41.30% | 54.70% | 59.96% | 61.75% | 63.41% |
| MAEPretrained | 51.51% | 56.16% | 62.31% | 67.08% | 84.32% | 27.78% | 31.48% | 37.04% | 37.04% | 29.63% | 9.47% | 12.21% | 11.84% | 13.30% | 12.67% |
| MLPMixerPretrained1k | 92.84% | 93.82% | 94.81% | 95.41% | 97.38% | 75.93% | 77.78% | 72.22% | 74.07% | 61.11% | 37.39% | 49.19% | 52.41% | 55.75% | 56.74% |
| MLPMixerPretrained21k | 95.42% | 95.33% | 95.62% | 96.52% | 97.55% | 83.33% | 81.48% | 75.93% | 75.93% | 77.78% | 48.30% | 64.37% | 66.79% | 70.15% | 70.84% |
| ResNet50Pretrained1k | 94.35% | 95.12% | 94.83% | 95.93% | 97.46% | 90.74% | 92.59% | 88.89% | 88.89% | 83.33% | 44.89% | 59.87% | 64.15% | 66.70% | 67.29% |
| ResNet50Pretrained21k | 95.20% | 96.40% | 95.89% | 96.65% | 97.67% | 81.48% | 79.63% | 77.78% | 77.78% | 75.93% | 51.21% | 65.75% | 69.47% | 72.01% | 73.90% |
| SimCLRPretrained | 95.50% | 95.98% | 96.14% | 96.45% | 97.52% | 83.33% | 83.33% | 81.48% | 77.78% | 72.22% | 44.33% | 59.82% | 65.39% | 67.93% | 68.93% |
| ViTPretrained1k | 95.72% | 96.42% | 96.27% | 96.57% | 97.95% | 94.44% | 88.89% | 88.89% | 87.04% | 77.78% | 50.62% | 64.09% | 67.14% | 72.33% | 73.19% |
| ViTPretrained21k | 94.91% | 95.22% | 96.09% | 96.45% | 97.71% | 83.33% | 83.33% | 83.33% | 83.33% | 77.78% | 49.36% | 64.21% | 67.42% | 70.77% | 72.13% |
| iBotPretrained1k | 96.42% | 96.58% | 96.60% | 97.40% | 97.28% | 88.89% | 83.33% | 87.04% | 81.48% | 79.63% | 44.58% | 62.59% | 68.10% | 71.20% | 73.36% |
| iBotPretrained21k | 96.16% | 96.14% | 96.38% | 97.37% | 97.27% | 88.89% | 90.74% | 90.74% | 83.33% | 81.48% | 51.20% | 68.58% | 71.57% | 74.62% | 75.94% |

Table A12: Background path linear eval top-1 accuracy across multiple percentages of varying training instances

| train_prop_to_vary | train_canonical_top_1_accuracy | | | | | val_canonical_top_1_accuracy | | | | | val_diverse_Translation_top_1_accuracy | | | | |
|---|---|---|---|---|---|---|---|---|---|---|---|---|---|---|---|
| model | 0.05 | 0.25 | 0.50 | 0.75 | 0.95 | 0.05 | 0.25 | 0.50 | 0.75 | 0.95 | 0.05 | 0.25 | 0.50 | 0.75 | 0.95 |
| CLIPPretrained | 97.49% | 97.00% | 97.25% | 97.55% | 96.80% | 87.04% | 90.74% | 77.78% | 81.48% | 75.93% | 45.33% | 69.74% | 74.53% | 77.72% | 78.19% |
| MAEPretrained | 96.75% | 96.63% | 97.20% | 97.46% | 97.91% | 83.33% | 74.07% | 66.67% | 64.81% | 72.22% | 29.17% | 51.35% | 61.05% | 65.16% | 70.07% |
| MLPMixerPretrained1k | 96.95% | 96.73% | 97.17% | 97.24% | 97.88% | 88.89% | 77.78% | 77.78% | 74.07% | 64.81% | 44.65% | 64.56% | 70.67% | 74.34% | 75.95% |
| MLPMixerPretrained21k | 97.71% | 97.69% | 97.90% | 97.98% | 98.35% | 85.19% | 88.89% | 85.19% | 81.48% | 77.78% | 46.53% | 70.54% | 77.27% | 79.78% | 81.68% |
| ResNet50Pretrained1k | 97.97% | 97.92% | 97.91% | 97.86% | 98.27% | 87.04% | 87.04% | 72.22% | 81.48% | 81.48% | 45.05% | 64.83% | 71.87% | 76.56% | 79.63% |
| ResNet50Pretrained21k | 97.54% | 97.62% | 97.74% | 97.69% | 98.20% | 88.89% | 83.33% | 83.33% | 83.33% | 77.78% | 52.22% | 70.96% | 75.78% | 81.10% | 82.45% |
| SimCLRPretrained | 97.40% | 97.57% | 97.68% | 97.81% | 98.12% | 90.74% | 77.78% | 77.78% | 83.33% | 77.78% | 45.21% | 68.73% | 74.59% | 76.55% | 79.86% |
| ViTPretrained1k | 97.80% | 97.88% | 98.00% | 97.92% | 98.28% | 90.74% | 90.74% | 85.19% | 81.48% | 81.48% | 49.30% | 71.82% | 76.95% | 80.39% | 82.58% |
| ViTPretrained21k | 97.80% | 97.59% | 97.89% | 97.91% | 98.25% | 87.04% | 88.89% | 81.48% | 81.48% | 87.04% | 45.83% | 71.80% | 75.51% | 79.96% | 81.86% |
| iBotPretrained1k | 97.77% | 97.55% | 97.64% | 97.77% | 98.06% | 88.89% | 85.19% | 79.63% | 75.93% | 79.63% | 45.69% | 68.94% | 74.76% | 76.63% | 79.83% |
| iBotPretrained21k | 97.97% | 97.83% | 97.88% | 97.92% | 98.13% | 88.89% | 87.04% | 88.89% | 81.48% | 79.63% | 49.84% | 70.97% | 78.13% | 82.53% | 84.07% |

Table A13: Position finetuning top-1 accuracy across multiple percentages of varying training instances

| train_prop_to_vary | train_canonical_top_1_accuracy | | | | | val_canonical_top_1_accuracy | | | | | val_diverse_Rotation_top_1_accuracy | | | | |
|---|---|---|---|---|---|---|---|---|---|---|---|---|---|---|---|
| model | 0.05 | 0.25 | 0.50 | 0.75 | 0.95 | 0.05 | 0.25 | 0.50 | 0.75 | 0.95 | 0.05 | 0.25 | 0.50 | 0.75 | 0.95 |
| CLIPPretrained | 97.60% | 97.01% | 97.30% | 97.58% | 97.97% | 88.89% | 88.89% | 85.19% | 83.33% | 72.22% | 34.29% | 57.01% | 62.67% | 65.58% | 68.45% |
| MAEPretrained | 96.87% | 96.75% | 97.25% | 97.52% | 97.95% | 75.93% | 68.52% | 62.96% | 68.52% | 62.96% | 22.64% | 45.24% | 50.87% | 54.18% | 53.82% |
| MLPMixerPretrained1k | 96.75% | 96.58% | 97.08% | 97.25% | 97.86% | 83.33% | 85.19% | 83.33% | 77.78% | 64.81% | 33.29% | 51.43% | 55.55% | 58.78% | 59.35% |
| MLPMixerPretrained21k | 97.68% | 97.65% | 97.90% | 97.90% | 98.35% | 87.04% | 87.04% | 77.78% | 77.78% | 75.93% | 38.93% | 62.76% | 67.97% | 71.90% | 73.47% |
| ResNet50Pretrained1k | 98.05% | 97.81% | 97.89% | 97.93% | 98.24% | 84.81% | 81.48% | 81.48% | 81.48% | 75.93% | 33.29% | 53.51% | 62.56% | 64.45% | 67.47% |
| ResNet50Pretrained21k | 97.52% | 97.45% | 97.65% | 97.61% | 98.18% | 85.19% | 79.63% | 83.33% | 87.04% | 85.19% | 37.45% | 59.85% | 65.39% | 70.66% | 73.13% |
| SimCLRPretrained | 97.37% | 97.43% | 97.53% | 97.74% | 98.07% | 87.04% | 87.04% | 83.33% | 87.04% | 75.93% | 35.50% | 55.87% | 64.34% | 66.79% | 69.34% |
| ViTPretrained1k | 97.94% | 97.87% | 97.98% | 97.95% | 98.31% | 88.89% | 87.04% | 85.19% | 77.78% | 75.93% | 40.13% | 62.66% | 68.53% | 71.32% | 73.36% |
| ViTPretrained21k | 97.68% | 97.61% | 97.90% | 97.97% | 98.27% | 88.89% | 79.63% | 83.33% | 74.07% | 70.74% | 39.75% | 61.42% | 68.39% | 71.51% | 73.76% |
| iBotPretrained1k | 97.49% | 97.55% | 97.56% | 97.75% | 98.02% | 88.89% | 77.78% | 83.33% | 75.93% | 68.52% | 36.30% | 60.69% | 65.98% | 68.22% | 70.00% |
| iBotPretrained21k | 98.00% | 97.79% | 97.96% | 98.01% | 98.19% | 88.89% | 87.04% | 83.33% | 85.19% | 72.22% | 38.86% | 62.35% | 69.18% | 71.96% | 73.84% |

Table A14: Pose finetuning top-1 accuracy across multiple percentages of varying training instances

| train_prop_to_vary | train_canonical_top_1_accuracy | | | | | val_canonical_top_1_accuracy | | | | | val_diverse_Spot hue_top_1_accuracy | | | | |
|---|---|---|---|---|---|---|---|---|---|---|---|---|---|---|---|
| model | 0.05 | 0.25 | 0.50 | 0.75 | 0.95 | 0.05 | 0.25 | 0.50 | 0.75 | 0.95 | 0.05 | 0.25 | 0.50 | 0.75 | 0.95 |
| CLIPPretrained | 97.66% | 97.13% | 97.40% | 97.62% | 97.98% | 90.74% | 88.89% | 87.04% | 87.04% | 81.48% | 49.99% | 70.06% | 75.12% | 82.36% | 82.48% |
| MAEPretrained | 97.04% | 96.67% | 97.20% | 97.41% | 97.95% | 79.63% | 77.78% | 72.22% | 74.07% | 68.52% | 30.10% | 54.63% | 60.74% | 66.78% | 69.34% |
| MLPMixerPretrained1k | 97.32% | 96.92% | 97.20% | 97.36% | 97.89% | 87.04% | 83.33% | 77.78% | 79.63% | 72.22% | 48.80% | 65.60% | 70.82% | 76.88% | 78.59% |
| MLPMixerPretrained21k | 97.80% | 97.83% | 97.99% | 97.99% | 98.42% | 88.89% | 87.04% | 81.48% | 75.93% | 79.63% | 49.81% | 73.15% | 75.23% | 79.01% | 82.35% |
| ResNet50Pretrained1k | 98.02% | 97.99% | 98.03% | 97.98% | 98.28% | 88.89% | 87.04% | 83.33% | 81.48% | 77.78% | 48.48% | 67.72% | 74.65% | 76.90% | 75.83% |
| ResNet50Pretrained21k | 97.68% | 97.60% | 97.87% | 97.79% | 98.24% | 90.74% | 87.04% | 83.33% | 77.78% | 75.93% | 54.35% | 71.69% | 76.70% | 81.03% | 81.86% |
| SimCLRPretrained | 97.60% | 97.61% | 97.72% | 97.87% | 98.17% | 90.00% | 76.30% | 85.19% | 77.78% | 74.07% | 45.58% | 70.36% | 79.09% | 78.06% | 81.75% |
| ViTPretrained1k | 97.68% | 97.93% | NaN | 98.00% | 98.37% | 94.44% | 88.89% | NaN | 81.48% | 81.48% | 54.38% | 70.94% | NaN | 82.53% | 83.94% |
| ViTPretrained21k | 97.77% | 97.68% | 97.94% | 98.00% | 98.24% | 92.59% | 85.19% | 77.78% | 78.15% | 81.48% | 52.49% | 72.55% | 76.58% | 79.75% | 82.39% |
| iBotPretrained1k | 97.91% | 97.58% | 97.76% | 97.88% | 98.08% | 88.89% | 81.48% | 79.63% | 75.93% | 74.07% | 48.67% | 69.24% | 75.01% | 79.44% | 80.81% |
| iBotPretrained21k | 98.25% | 97.87% | 97.88% | 97.96% | 98.13% | 90.74% | 83.33% | 92.59% | 79.63% | 83.33% | 49.39% | 74.42% | 80.67% | 83.05% | 85.02% |

Table A15: Spot hue finetuning top-1 accuracy across multiple percentages of varying training instances

| train_prop_to_vary | train_canonical_top_1_accuracy | | | | | val_canonical_top_1_accuracy | | | | | val_diverse_Scale_top_1_accuracy | | | | |
|---|---|---|---|---|---|---|---|---|---|---|---|---|---|---|---|
| model | 0.05 | 0.25 | 0.50 | 0.75 | 0.95 | 0.05 | 0.25 | 0.50 | 0.75 | 0.95 | 0.05 | 0.25 | 0.50 | 0.75 | 0.95 |
| CLIPPretrained | 97.68% | 97.08% | 97.39% | 97.63% | 97.95% | 90.74% | 90.74% | 87.04% | 83.33% | 79.63% | 45.47% | 66.39% | 72.46% | 75.63% | 78.05% |
| MAEPretrained | 96.81% | 96.64% | 97.19% | 97.40% | 97.91% | 81.48% | 70.37% | 70.37% | 70.37% | 53.70% | 28.70% | 51.73% | 60.66% | 62.49% | 64.47% |
| MLPMixerPretrained1k | 97.23% | 96.60% | 97.07% | 97.24% | 97.82% | 87.04% | 85.19% | 83.33% | 74.07% | 70.37% | 41.62% | 60.01% | 65.78% | 70.24% | 71.64% |
| MLPMixerPretrained21k | 97.80% | 97.80% | 97.96% | 97.95% | 98.37% | 85.19% | 87.04% | 81.48% | 72.22% | 77.78% | 46.02% | 68.37% | 73.08% | 76.51% | 77.38% |
| ResNet50Pretrained1k | 97.88% | 97.94% | 97.95% | 97.89% | 98.24% | 87.04% | 85.19% | 81.48% | 75.93% | 79.63% | 44.47% | 62.86% | 71.55% | 73.81% | 75.80% |
| ResNet50Pretrained21k | 97.40% | 97.58% | 97.84% | 97.72% | 98.18% | 90.74% | 79.63% | 81.48% | 79.63% | 79.63% | 49.37% | 68.72% | 70.89% | 76.85% | 79.18% |
| SimCLRPretrained | 97.57% | 97.54% | 97.65% | 97.83% | 98.10% | 88.89% | 83.33% | 83.33% | 83.33% | 74.44% | 42.25% | 65.04% | 71.88% | 74.89% | 76.25% |
| ViTPretrained1k | 97.80% | 97.92% | 98.05% | 97.92% | 98.34% | 88.89% | 83.33% | 85.19% | 79.63% | 79.63% | 44.17% | 65.86% | 71.66% | 77.46% | 78.46% |
| ViTPretrained21k | 97.77% | 97.71% | 97.85% | 97.99% | 98.24% | 85.19% | 85.19% | 85.19% | 79.63% | 79.63% | 42.63% | 67.71% | 70.61% | 74.96% | 76.14% |
| iBotPretrained1k | 97.85% | 97.61% | 97.74% | 97.79% | 98.04% | 90.74% | 87.04% | 83.33% | 83.33% | 79.63% | 45.14% | 64.09% | 71.34% | 73.90% | 76.99% |
| iBotPretrained21k | 97.88% | 97.75% | 97.86% | 97.97% | 98.12% | 88.89% | 87.04% | 87.04% | 81.48% | 75.93% | 48.70% | 65.52% | 72.39% | 79.16% | 80.04% |

Table A16: Scale finetuning top-1 accuracy across multiple percentages of varying training instances

| train_prop_to_vary | train_canonical_top_1_accuracy | | | | | val_canonical_top_1_accuracy | | | | | val_diverse_Background path_top_1_accuracy | | | | |
|---|---|---|---|---|---|---|---|---|---|---|---|---|---|---|---|
| model | 0.05 | 0.25 | 0.50 | 0.75 | 0.95 | 0.05 | 0.25 | 0.50 | 0.75 | 0.95 | 0.05 | 0.25 | 0.50 | 0.75 | 0.95 |
| CLIPPretrained | 96.49% | 97.05% | 97.52% | 97.81% | 98.01% | 94.44% | 88.89% | 92.59% | 87.04% | 81.48% | 50.79% | 67.24% | 76.25% | 79.79% | 80.92% |
| MAEPretrained | 96.20% | 96.61% | 97.31% | 97.63% | 97.99% | 81.85% | 72.22% | 77.78% | 72.22% | 62.96% | 30.49% | 52.04% | 64.77% | 66.67% | 68.56% |
| MLPMixerPretrained1k | 96.16% | 97.03% | 97.13% | 97.66% | 97.76% | 90.74% | 87.04% | 81.48% | 75.93% | 72.22% | 47.41% | 63.84% | 71.85% | 73.76% | 75.96% |
| MLPMixerPretrained21k | 98.08% | 98.15% | 98.09% | 98.33% | 98.71% | 88.89% | 92.59% | 90.74% | 85.19% | 79.63% | 51.84% | 72.00% | 77.05% | 80.46% | 81.10% |
| ResNet50Pretrained1k | 97.71% | 97.76% | 97.95% | 98.04% | 98.38% | 92.59% | 92.59% | 90.74% | 92.59% | 83.33% | 46.50% | 63.48% | 70.90% | 73.73% | 77.72% |
| ResNet50Pretrained21k | 97.38% | 98.09% | 97.72% | 98.33% | 98.34% | 88.89% | 83.33% | 87.04% | 81.48% | 85.19% | 50.87% | 71.59% | 76.21% | 79.13% | 83.24% |
| SimCLRPretrained | 97.38% | 97.31% | 97.77% | 97.55% | 98.05% | 77.78% | 87.04% | 88.89% | 90.74% | 83.33% | 43.32% | 61.47% | 69.94% | 76.99% | 79.07% |
| ViTPretrained1k | 98.12% | 97.76% | 97.89% | 97.73% | 98.44% | 88.89% | 90.74% | 87.04% | 81.48% | 83.33% | 54.25% | 71.56% | 75.73% | 80.58% | 84.92% |
| ViTPretrained21k | 97.77% | 97.49% | 97.95% | 98.04% | 98.37% | 91.67% | 88.89% | 90.74% | 87.04% | 81.48% | 55.17% | 73.53% | 76.50% | 82.28% | 81.94% |
| iBotPretrained1k | 97.47% | 97.44% | 97.61% | 98.15% | 97.86% | 88.89% | 92.59% | 90.74% | 81.48% | 79.63% | 51.19% | 71.17% | 75.32% | 78.37% | 79.93% |
| iBotPretrained21k | 97.69% | 97.91% | 97.77% | 98.17% | 97.93% | 93.52% | 92.59% | 90.74% | 85.19% | 83.33% | 55.00% | 73.11% | 76.62% | 83.56% | 82.90% |

Table A17: Background path finetuning top-1 accuracy across multiple percentages of varying training instances

## J   All Factor Gaps

We also study the setting where all factors vary during training. In Figures A21 and A22 we show the generalization gaps when all factors vary for linear evaluation and finetuning.

## K   Class generalization

In addition to the finetning results, we include here linear evaluation results for class generalization gaps A18. We show generalization gaps across classes by comparing the similarity of classes seen during training to held-out classes in Figure A23.

## L   Cross factor effects

We study the effect of varying a factor on the generalization gaps of other factors. In Figures A24 and A25 we show the slopes of the generalization gaps as the number of varying training instances increases during training. We see how varying one factor can also close the robustenss gap of other factors. We also show normalized versions of these plots in A26 and A27.

| model | Position gap | Pose gap | Lighting color gap | Size gap | Background gap | Average gap |
|---|---|---|---|---|---|---|
| CLIP | -41.69 | -46.51 | -43.73 | -41.51 | -48.09 | -44.31 |
| MAE | -3.32 | -6.64 | -7.69 | -5.68 | -14.61 | -7.59 |
| MLPMixer1k | -39.07 | -43.70 | -43.42 | -36.32 | -47.79 | -42.06 |
| MLPMixer21k | -43.07 | -57.34 | -42.90 | -44.84 | -52.09 | -48.05 |
| ResNet50-1k | -44.97 | -51.06 | -43.84 | -42.18 | -55.67 | -47.54 |
| ResNet50-21k | -46.34 | -56.89 | -44.06 | -47.26 | -57.81 | -50.47 |
| ViT-1k | -47.03 | -58.69 | -45.24 | -48.67 | -52.77 | -50.48 |
| ViT-21k | -50.41 | -56.94 | -49.17 | -49.76 | -55.23 | -52.30 |
| iBot-1k | -46.90 | -61.71 | -46.35 | -51.56 | -59.88 | -53.28 |
| iBot-21k | -53.00 | -64.94 | -50.99 | -56.05 | -64.83 | -57.96 |
| Average | -41.58 | -50.44 | -41.74 | -42.38 | -50.88 | -45.40 |

Table A18: Linear eval class generalization top-1 accuracy gaps: shows validation top-1 accuracy difference between classes (27 randomly selected) seen with diversity and those not.

## M  Effect of class similarity on models' ability to generalize variation across classes

We study the effect of class similarity by measuring the generalization gaps per class for each factor relative to the class's similarity to the nearest class seen varying during training. If models' are able to generalize variation across classes, we might expect models generalize variation better when the class is similar to one seen varying during training. In Figures A28, A28, A32, and A30.

## N  Experiments details

Tables 1a and 1b show results for the best model after 10k steps of training with adam on 6 log scale learning rates (1e-2 to 1e-6) cross validated on canonical top-1 accuracy for validation images.

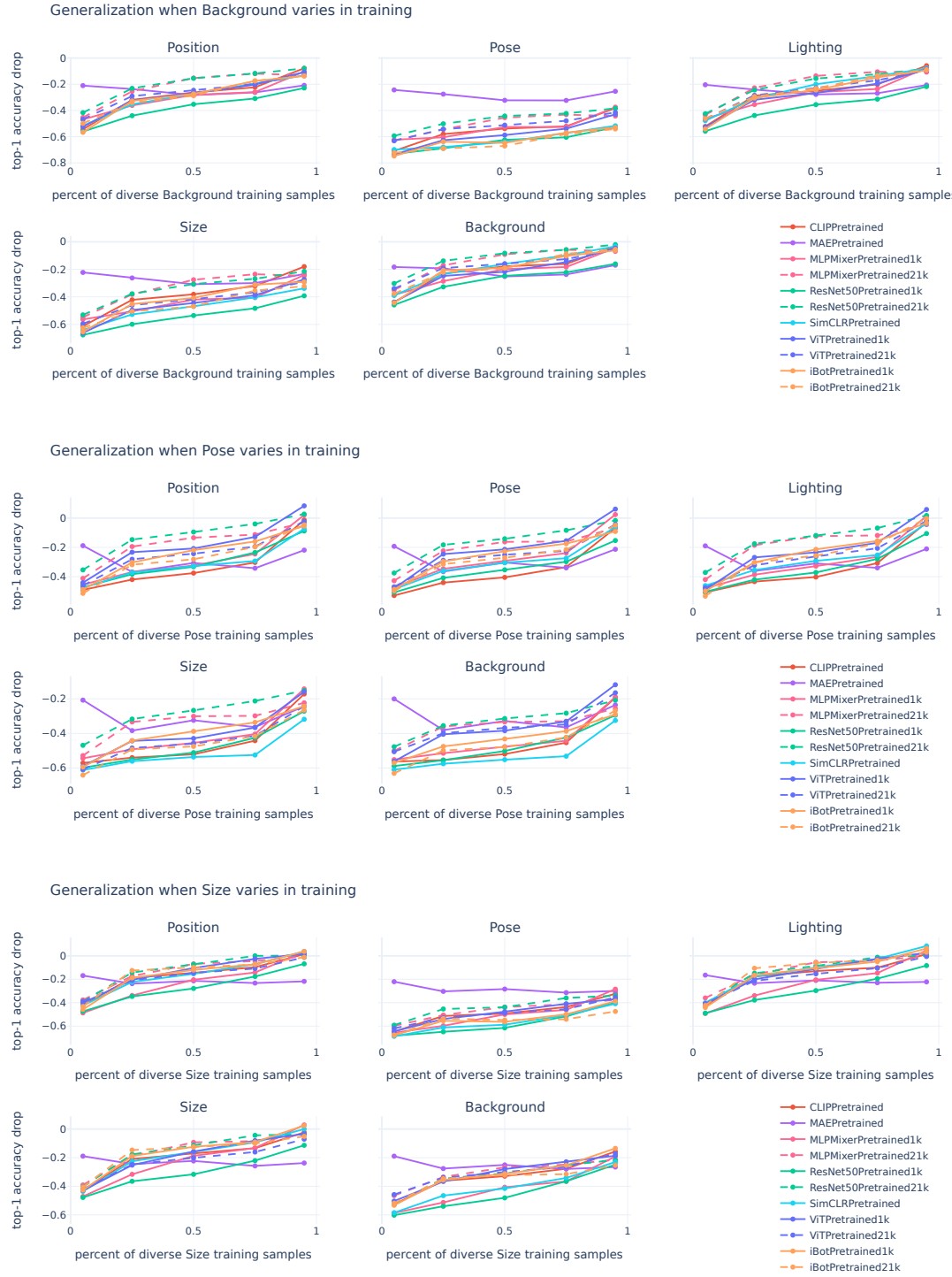

Figure A17: Linear Evaluation Effect of Variability in Training (part 1)

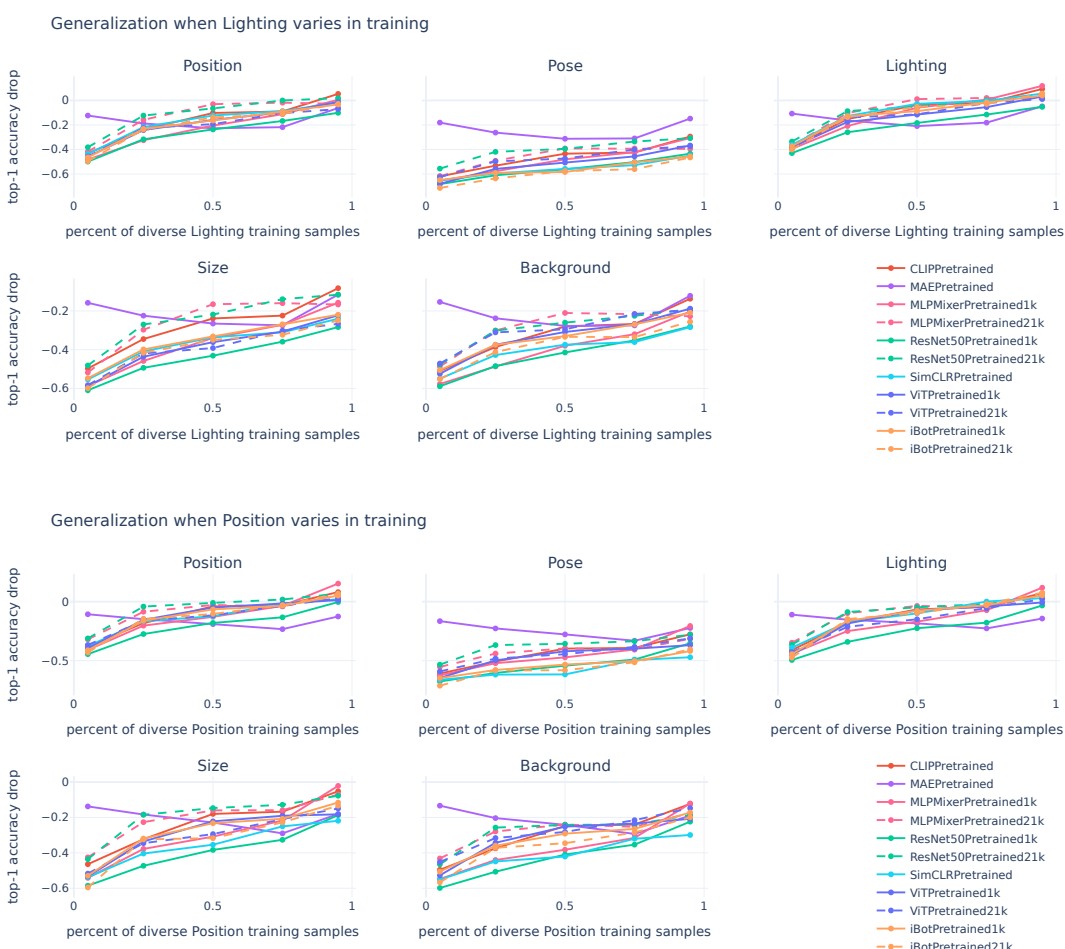

Figure A18: Linear Evaluation Effect of Variability in Training (part 2)

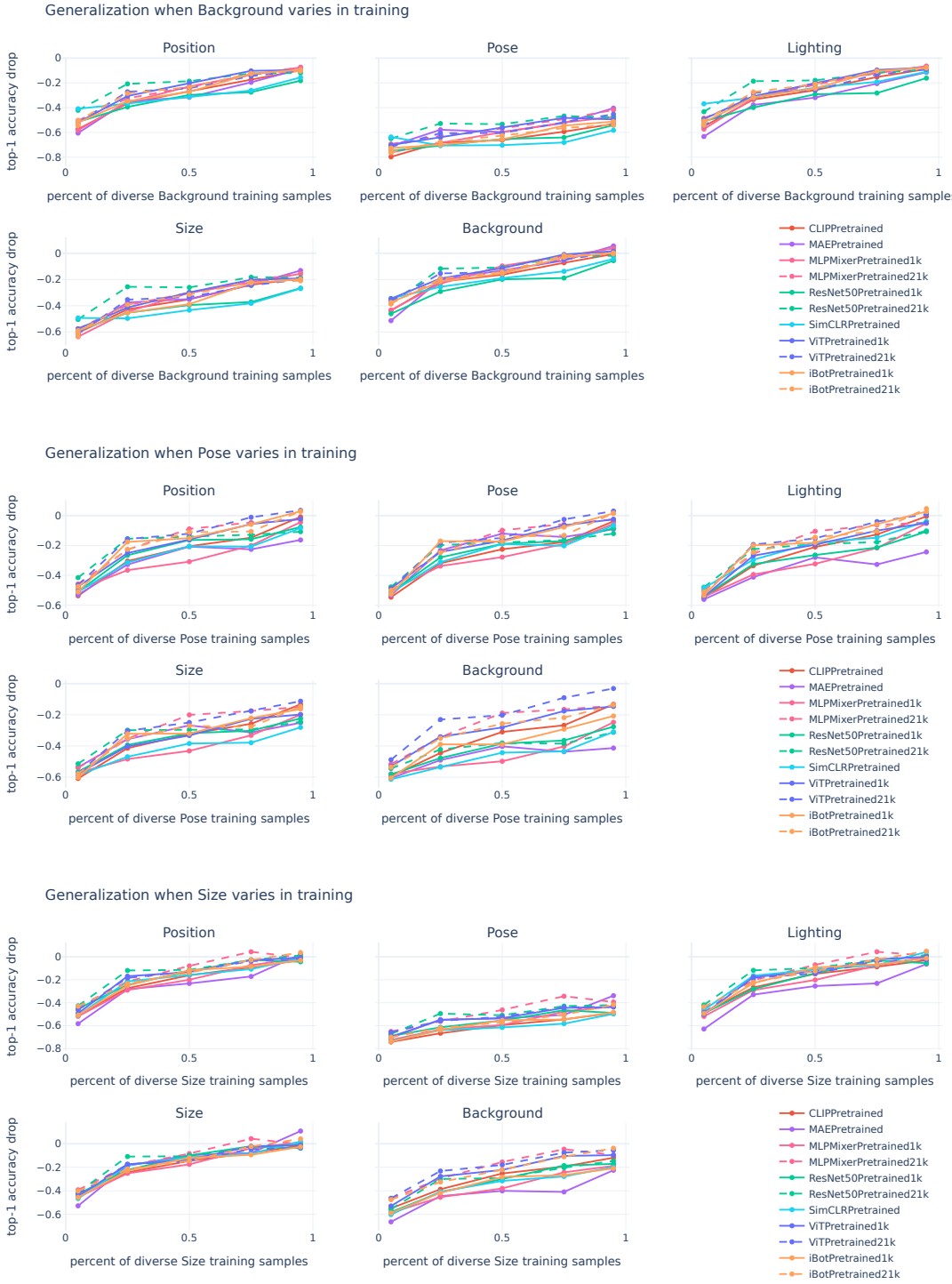

Figure A19: Finetuning Effect of Variability in Training (part 1)

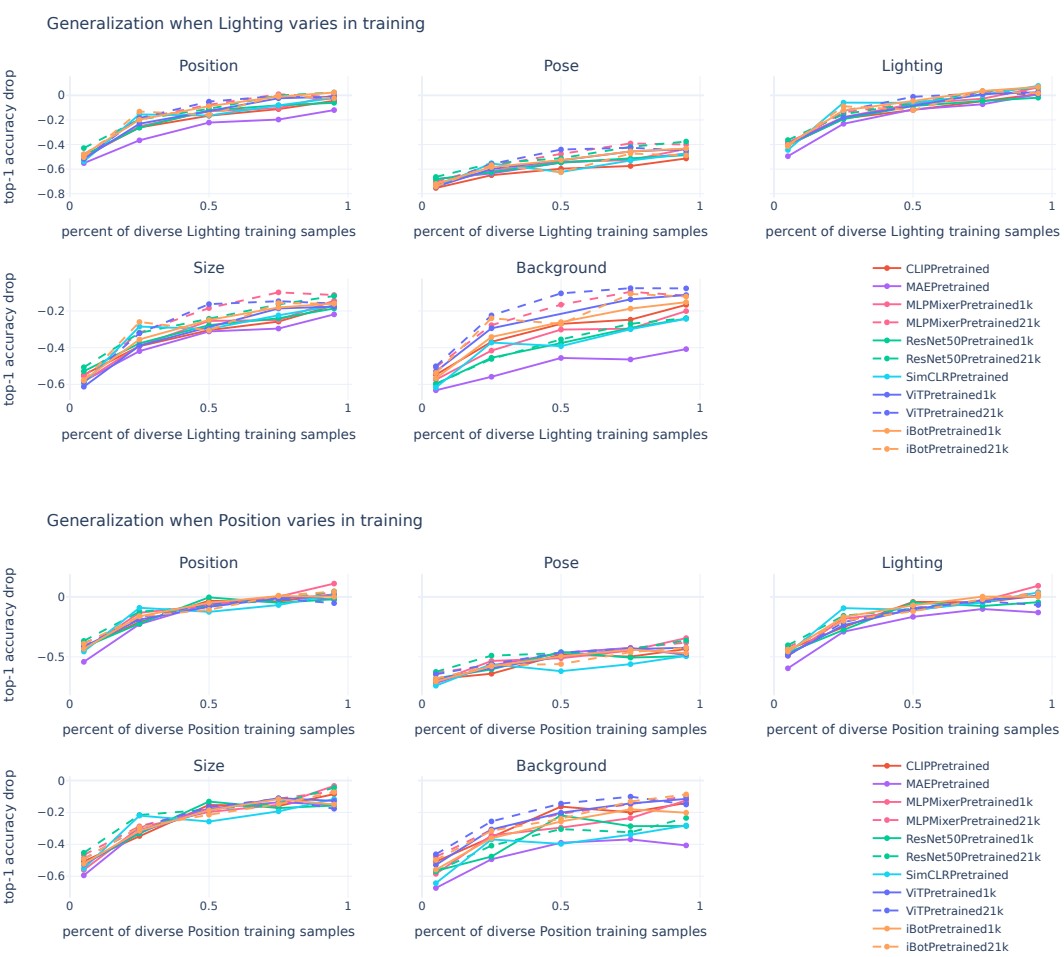

Figure A20: Finetuning Effect of Variability in Training (part 2)

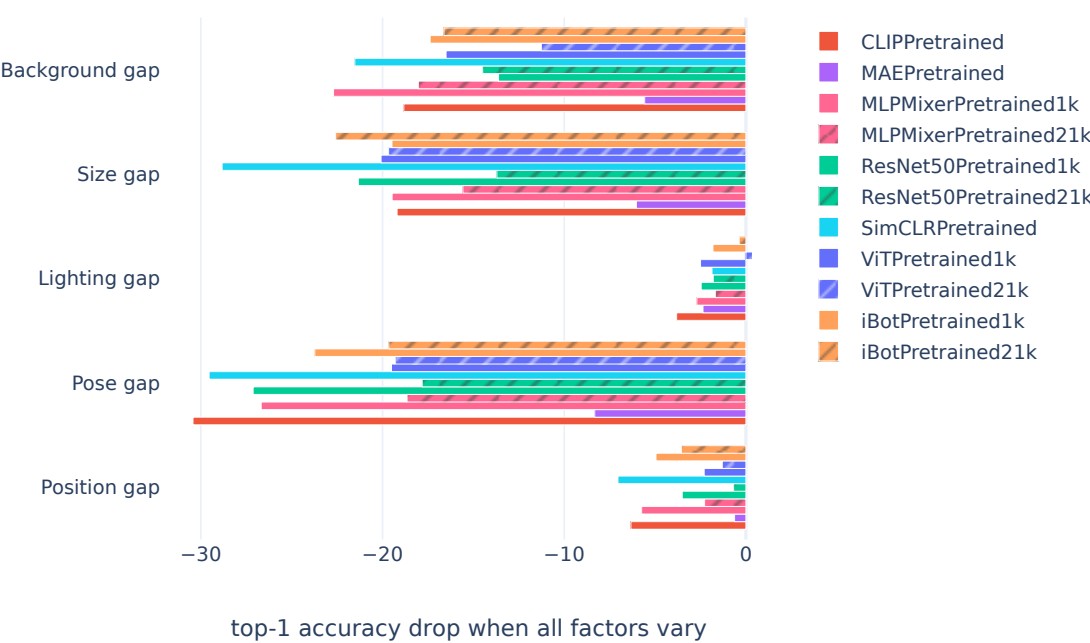

Figure A21: Generalization gaps when all factors vary during training with linear evaluation

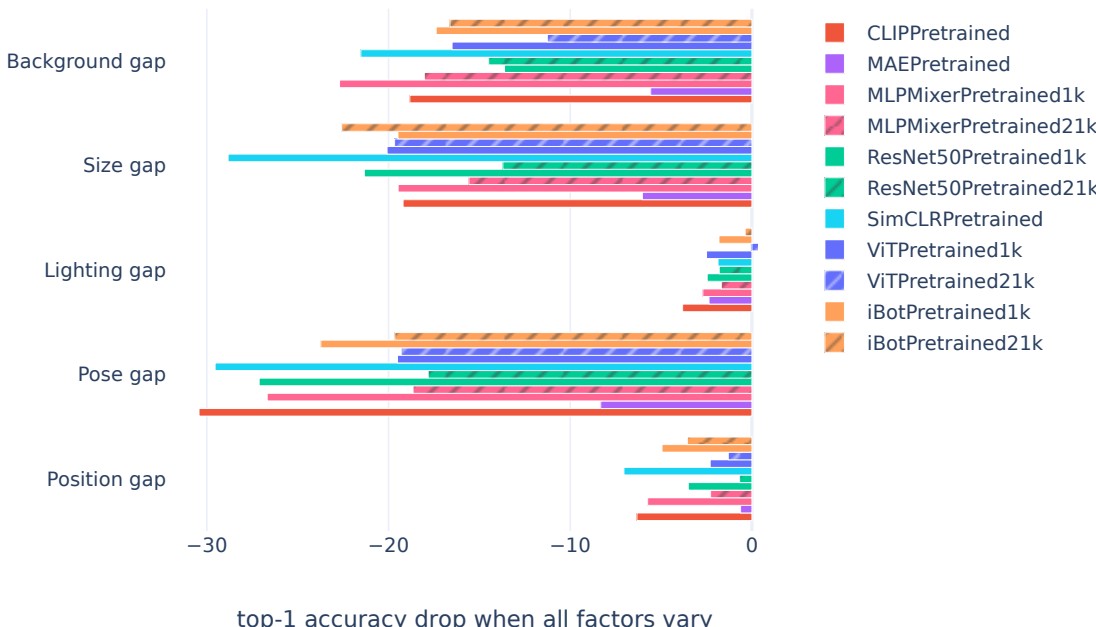

Figure A22: Generalization gaps when all factors vary during training with finetuning

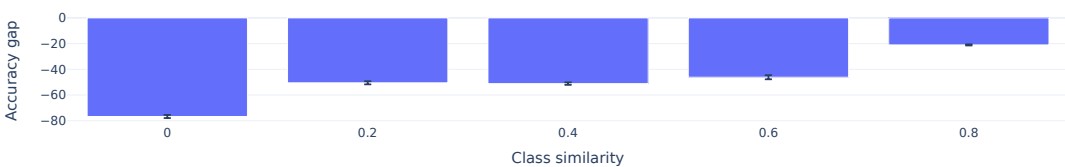

Figure A23: **Generalization gaps are smaller only for classes very similar to those seen during training and worse for classes that are very dissimilar.** We plot the generalization gaps as similarity to the nearest class seen varying during training increases using the mean accuracy gap with error bars indicating the standard error.

| varying factor | Position gap | Pose gap | Lighting gap | Size gap | Background gap |
|---|---|---|---|---|---|
| Position | 0.385894 | 0.249188 | 0.411196 | 0.350276 | 0.287304 |
| Pose | 0.383425 | 0.369503 | 0.400835 | 0.308550 | 0.280720 |
| Lighting | 0.398418 | 0.243639 | 0.385000 | 0.330450 | 0.281797 |
| Size | 0.385311 | 0.245047 | 0.395171 | 0.352747 | 0.294040 |
| Background | 0.343635 | 0.209684 | 0.354958 | 0.304687 | 0.300778 |

Figure A24: Spill over effects: shows the average slope across models when a given factor varies during linear evaluation

| varying factor | Position gap | Pose gap | Lighting gap | Size gap | Background gap |
|---|---|---|---|---|---|
| Position | 0.448903 | 0.262276 | 0.475593 | 0.419783 | 0.370703 |
| Pose | 0.442200 | 0.454216 | 0.469521 | 0.372108 | 0.353449 |
| Lighting | 0.480997 | 0.277407 | 0.444130 | 0.415490 | 0.384728 |
| Size | 0.483358 | 0.274840 | 0.487531 | 0.451954 | 0.427255 |
| Background | 0.425475 | 0.242605 | 0.437238 | 0.406613 | 0.405921 |

Figure A25: Spill over effects: shows the average slope across models when a given factor varies for finetuning

| varying factor | Position gap | Pose gap | Lighting gap | Size gap | Background gap |
|---|---|---|---|---|---|
| Position | 1.000000 | 0.674386 | 1.068042 | 0.992995 | 0.955206 |
| Pose | 0.993602 | 1.000000 | 1.041130 | 0.874707 | 0.933316 |
| Lighting | 1.032454 | 0.659368 | 1.000000 | 0.936791 | 0.936894 |
| Size | 0.998488 | 0.663178 | 1.026418 | 1.000000 | 0.977599 |
| Background | 0.890490 | 0.567474 | 0.921968 | 0.863755 | 1.000000 |

Figure A26: Normalized Spill over effects: shows the average slope across models when a given factor varies during linear evaluation. Normalization is across rows by dividing the diagonal value to isolate how much more a given spill-over effect than the intended.

| varying factor | Position gap | Pose gap | Lighting gap | Size gap | Background gap |
|---|---|---|---|---|---|
| Position | 1.000000 | 0.577425 | 1.070841 | 0.928819 | 0.913239 |
| Pose | 0.985068 | 1.000000 | 1.057171 | 0.823333 | 0.870733 |
| Lighting | 1.071495 | 0.610738 | 1.000000 | 0.919320 | 0.947789 |
| Size | 1.076755 | 0.605087 | 1.097721 | 1.000000 | 1.052556 |
| Background | 0.947811 | 0.534118 | 0.984481 | 0.899679 | 1.000000 |

Figure A27: Normalized Spill over effects: shows the average slope across models when a given factor varies for finetuning. Normalization is across rows by dividing the diagonal value to isolate how much more a given spill-over effect than the intended.

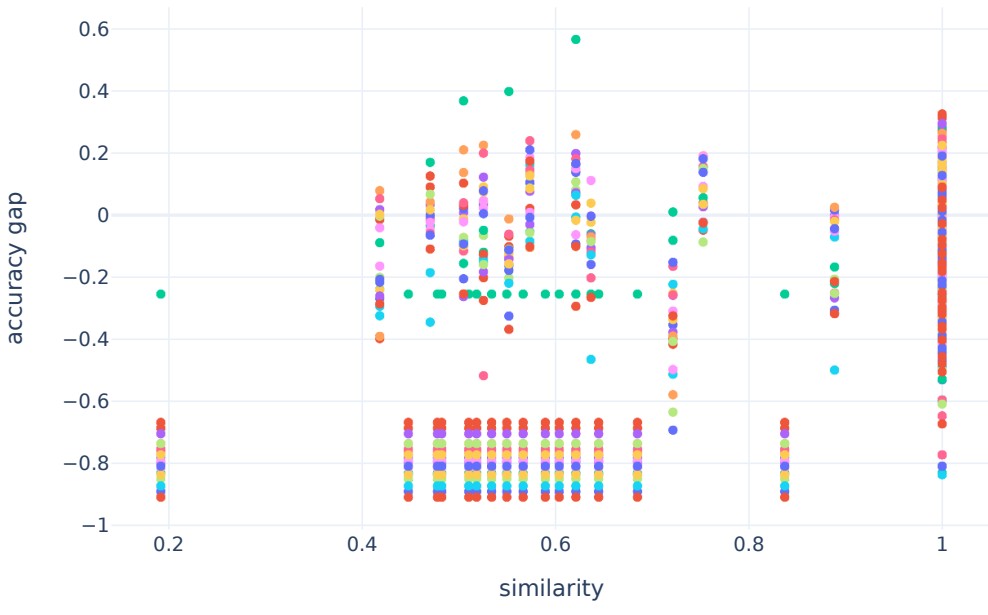

Figure A28: Position gap as class similarity to nearest neighbor increases to classes seen varying during training.

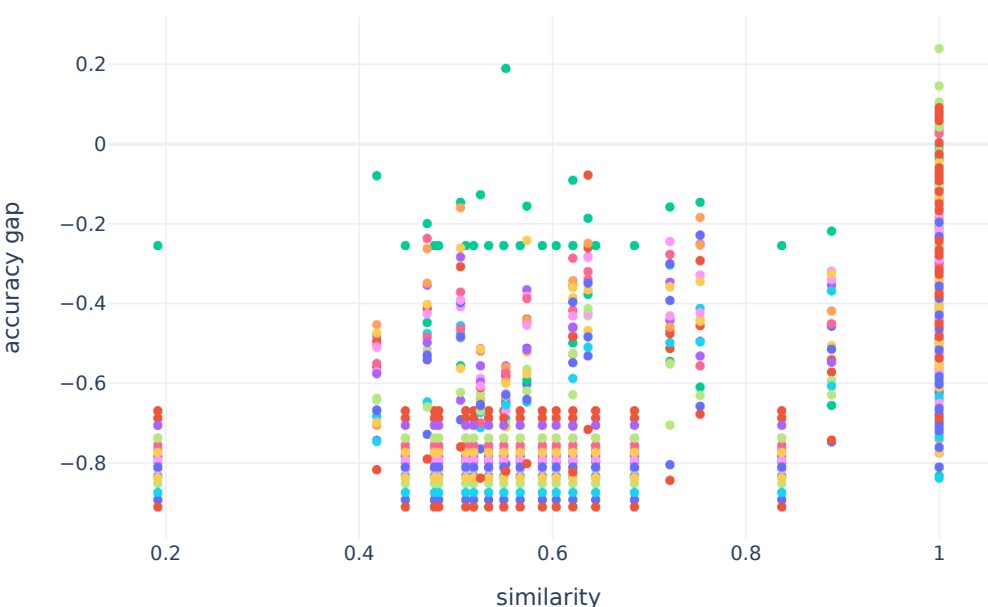

Figure A29: Pose gap as class similarity to nearest neighbor increases to classes seen varying during training.

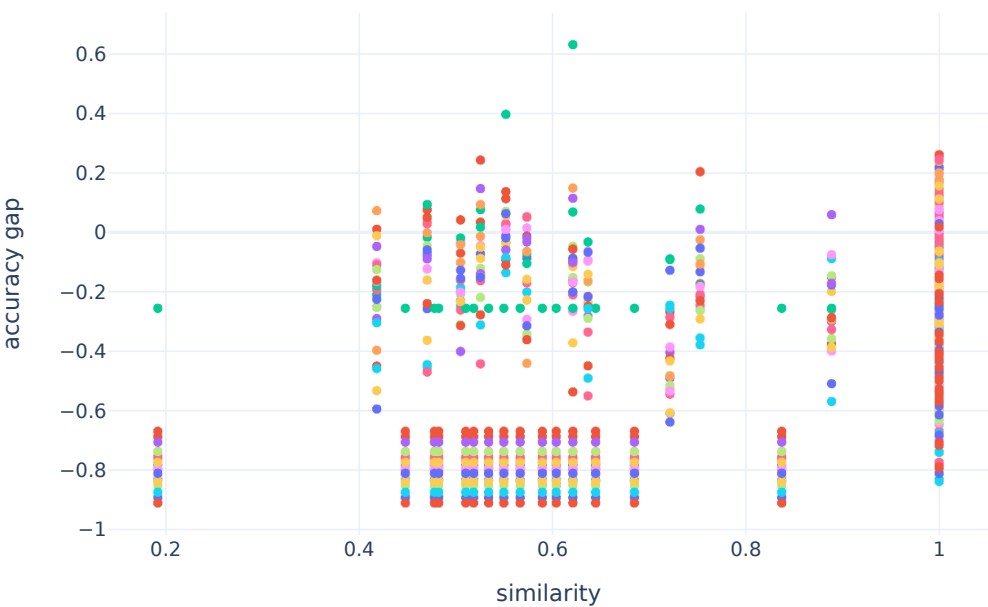

Figure A30: Scale gap as class similarity to nearest neighbor increases to classes seen varying during training.

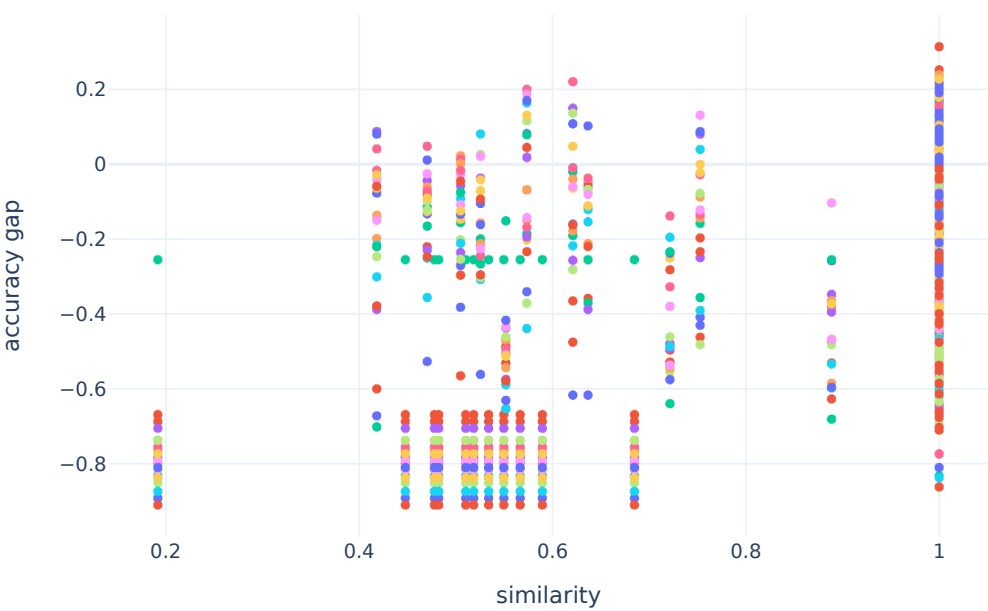

Figure A31: Background gap as class similarity to nearest neighbor increases to classes seen varying during training.

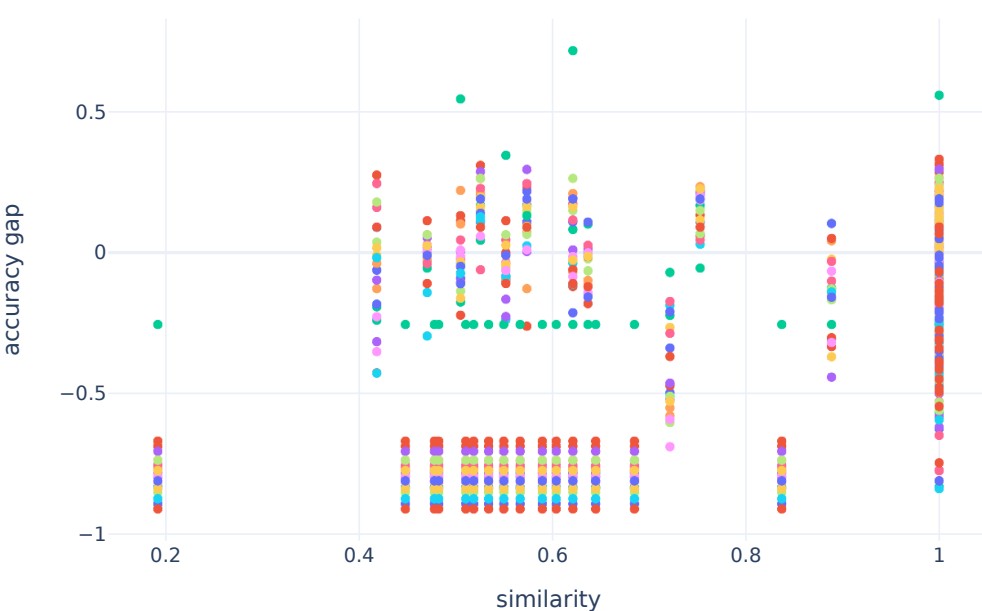

Figure A32: Lighting color gap as class similarity to nearest neighbor increases to classes seen varying during training.

