# OpenReview forum: "The Robustness Limits of SoTA Vision Models to Natural Variation"
_TMLR — Accepted by TMLR_

### Review · Reviewer_AUUo · 2023-01-19

**Summary Of Contributions:**

This paper describes a controlled study of robustness in generalization to common variation factors for objects --- in particular, pose, position, background, lighting and size (relative to background) --- in the context of transfer learning.  3D models were rendered against pre-selected backgrounds in different configurations of these variables, with one configuration selected as the "canonical" representation used when not applying variations.  The key metric studied is the difference in average classification accuracy between images with varying factors, versus images with only the canonical values ("generalization gap").  Different training-time settings are compared, to study how training variation affects test-time generalization.  Training setups that use all variations, no variations, or variations of only some factors but not others are evaluated and compared in order to characterize generalization abilities, as are experiments on pretraining data and model architecture.  Several conclusions are drawn or confirmed, including (a) larger/diverse pretraining (imnet21k vs 1k) helps generalization, (b) more diverse training samples helps generalization not only in the factors varied, but sometimes others as well (however, see question below on this), (c) most current classification architectures show behavior in generalization gaps, (d) improving generalization on one class does not usually improve generalization on other classes.

**Audience:**

Yes

**Claims And Evidence:**

Yes

**Requested Changes:**

* discussion/comparison relative to Kumar &al https://arxiv.org/pdf/2202.10054.pdf

* clearer descriptions of data sampling, transfer training and experimental procedures

Both of the above are necessary changes

**Strengths And Weaknesses:**

This is a thorough set of experiments, confirming the observed behavior across a wide range of architectures and pretraining methods, and evaluating multiple controlled transfer training conditions.  Some interesting observations are presented, in particular the measurements showing the rates at which generalization improves when increasing variation (how much diversity is needed to improve generalization?), and the effect varying one factor has on other factors.  The effects of pretraining dataset (larger is more robust) and architecture (most are similar behavior) are more widely recognized, but this offers further corroboration of these as well.

At the moment, however, I found many of the descriptions of data creation and evaluations vague, and details on the experimental setups are missing.  This makes the paper a somewhat hard to follow, as I was guessing at a lot.  More concrete figures and examples describing the data sampling and experimental setups would help.

In particular:

* I think it could be clearer including both absolute performance numbers as well as the "gap" measurement.  Currently, the size of the gaps is not put in context relative to the base point.  Plain accuracy numbers are reported here and there in the text, but it would help to be able to view and study these along with the relative drops.

* More precisely defining measurements and definition of "gap" might help as well.  While a very formal description in set notation is probably not needed, an algorithmic or mathematical description that more precisely includes which objects and factors are used in which averages would be helpful.

* The training procedure should be more concretely defined.  While the overall setting of linear classifier or end-to-end fine tuning is clear, exact setup and details are not.  Most importantly, the data sampling procedure is not very clear yet.  How are objects selected, and how many?  How is the canonical factor value assigned?  How long is each model trained (how many steps/samples) and how does this compare to the number of unique samples in the new training sets?  In particular, does adding or removing variation affect the number of training samples?  Algorithmic description and example images in figures describing what values/images are sampled in each experimental setting would help.

* What image augmentations were applied at training time --- just color transforms and translation?  Or also scale, in-plane rotation, or others?  Some of these might actually approximate more of the 3d transforms and change the amount of robustness.  Is lack of robustness from object size changes even after similarly sized scale augmentations are applied to the images?  How much does this using vs not using this planar augmentation change robustness?



The paper is also very related to Kumar &al 2022 https://arxiv.org/pdf/2202.10054.pdf , which analyzes out-of-distribution robustness in the context of transfer learning with linear, end-to-end, and hybrid trained classifiers.  I think this paper should include a discussion on how these work relate, and relations between their conclusions.


More detailed questions:

Entanglement of position and lighting:  In fact there is an entanglement created by 3D rendering.  When the object moves in world space and the light source remains fixed, this has the effect of (a) moving the object in the image plane, (b) moving the object relative to the light source, and (c) changing the result of perspective deformations by moving the object in the camera frustum.  I also noticed in the appendix images, that changing position seems to increase the size of the object between right and left.  Do these relations in the rendering pipeline account for the entanglement observed in performance robustness between factors?

Fig 1:  "chance seen varying during training" and "half seen varying during training":  I'm not sure what exactly these mean --- what is the selection procedure, and where does this chance fit in to this?  For example, is 50% the fraction of objects where all values are varied?  Or the fraction of times a random value is selected vs the canonical value, for all objects?  Or is it used in another way?  The use of "varying" e.g. in sec 5.1 is vague still in the text --- what is the sampling procedure?

Sec 5.2:  Fig 4 vs Fig A14:  The lines in A14 all trend upward, and don't clearly show the behavior described in the text where performance is worse than with zero variation.  To clearly demonstrate this, it would be best for the plot to show a downward dip followed by upward recovery.  In particular, if results at 0% variation are included, and a few more samples between 0% and 5% (e.g., 1%, 2%) that could show the down-up behavior.  In addition, the 0% values should coincide with the values for no variation in Table 1.

Figures 2, 3:  Captions or text should explain what ranges are represented with box plots (mean, 50 and 90 percentiles?  or other ranges?)

---

> ### Author Response · Authors · 2023-02-17
> **Thank you for the suggestions**
>
> We’re very glad you found our experiments thorough in terms of both models and settings, our study of generalization interesting, and conclusions noteworthy. We thank you for pointing out ways to clarify our experimental setup. We’ve added a new detailed description of the data generation, experimental settings along with new figures that we believe greatly improve clarity. We’d be happy to incorporate any further suggestions. We discuss the specific points below:
>
>
> > “I think it could be clearer including both absolute performance numbers as well as the "gap" measurement”
>
> We agree presenting absolute performance could aid readers in interpreting the performance gaps we measure. We now include in a new Appendix C tables showing both absolute accuracy as well as the relative gap for each model both for linear evaluation and finetuning protocols.
>
> > “More precisely defining measurements and definition of "gap" might help as well.”
>
> We apologize for the oversight and agree an explicit definition would be helpful. We now include an explicit definition at the start of Section 4  of the performance gap in terms of the drop in accuracy compared to models’ performance on canonical samples.
>
> > The training procedure should be more concretely defined…How are objects selected, and how many? How is the canonical factor value assigned?
>
>
> We agree clearly detailing the training setup for each setting is important. We now highlight at the beginning of each experimental section an explicit discussion of the data sampling and training details (changes highlighted in blue). We note our experiments explore several generalization settings: 1) out-of-the-box generalization Section 4, 2) training with variability of factors across all instances in Section 5.1, 3) training with variability of factors only for some instances (“how well do models generalize variation across instances”) Section 5.2, 4) training with variability of factors only for some classes (“how well do models generalize across classes”). We apply the same training setup for all models specifically we perform linear evaluation or finetuning for 10k steps, sweep over 6 learning rates between 1e-2 and 1e-6 (log scale) with the Adam optimizer.
> In all our setups, the objects were selected using the same curation (in terms of synsets) as ShapeNetCore, giving us a wide variety of quality models. For the canonical values, we considered either 0 transformation when possible (object at the original scale, at the origin, facing forward) since the models are aligned. For other parameters we chose arbitrary, yet sensible values. In the case of the background we opted for an image of the sky, and for the lighting a slightly yellowish hue, similar to what a lamp would give.
> We have added more details on the generation process in the appendix section B.1.

---

> > ### Author Response · Authors · 2023-02-17
> > **Continued responses**
> >
> > > What image augmentations were applied at training time --- just color transforms and translation? Or also scale, in-plane rotation, or others? Some of these might actually approximate more of the 3d transforms and change the amount of robustness
> >
> > All of the models are pre-trained using the standard data-augmentation pipeline proposed for each method. We load pretrained backbones for each model without modifying the pretraining data augmentations. For example, standard augmentation in pre-training for SimCLR consists of color jitter, random crops, and random Gaussian blur. For ResNet, we load pretrained models using the “ResNet Strikes Back” recipe, which includes random cropping, horizontal flipping, Mixup, CutMix. Of course, not all augmentations are used for every method, for example CutMix and mixup are only used for supervised models, but they should give an idea of the transformations used in pretraining. We’ve included an additional description to clarify this in Section 3.2.
> >
> > In our training protocols for both finetuning and linear evaluation , we apply only random resized cropping and horizontal flip, the standard augmentations commonly used in evaluating backbone representations.
> >
> > These transformations can approximate some of our "3d transforms", for example cropping can approximate an increase of the objects' size (a constrained kind of scaling) or translations, color transformations can approximate the change in lighting color. However more complex "3d transforms" such as rotation or change of the background are not related to augmentations used during the pretraining of the models.
> >
> > Looking beyond data augmentation, some variability is also present in the training data, where objects are seen in various scenarios (background, pose, lighting) and so the impact of data augmentation must be interpreted carefully.
> > Nonetheless we see that when changing factors of variations that are related to augmentations seen during training such as the lighting color, the drops in performance are smaller in general. We still see that changing the pose, which is related to cropping, leads to large drops in accuracy (see figure 1).
> >
> > > Entanglement of position and lighting
> >
> > Thank you for pointing this out. Most, if not all, factors of variation are correlated, and cannot perfectly be disentangled. Position and lighting is clearly visible, but position also affects scale (and vice versa), and position also changes what is seen from the background. These correlations are similar to what we would observe in the real world with an object moving in a scene or with certain components of it evolving.
> >
> > In practice, we do see that this entanglement is not severe (it mostly affects extreme cases) when looking at the performance drop for position and scale in figure 1. There is a drop of 6.4 points for position and 20.2 points for scale, showing very different behaviors. This discrepancy would indicate that this is not a problem and does not limit the conclusions that we can obtain from our experimental setup.
> >
> > > Fig 1: "chance seen varying during training" and "half seen varying during training": I'm not sure what exactly these mean --- what is the selection procedure, and where does this chance fit in to this?
> >
> > We now include more detailed descriptions of the sampling procedure for both instances and factor changes at the start of each section and in figure captions to ensure the setup is clear. We also provide an example to illustrate this sampling intuitively. We welcome additional suggestions to improve clarity.
> >
> > > Sec 5.2: Fig 4 vs Fig A14: The lines in A14 all trend upward, and don't clearly show the behavior described in the text where performance is worse than with zero variation. To clearly demonstrate this, it would be best for the plot to show a downward dip followed by upward recovery. In particular, if results at 0% variation are included, and a few more samples between 0% and 5% (e.g., 1%, 2%) that could show the down-up behavior. In addition, the 0% values should coincide with the values for no variation in Table 1.
> >
> > The results in Fig. 4 and Fig A14 depict results for different settings and are therefore not directly comparable. In Fig. 4 we test how well models generalize variation when the factor varies across all instances. During training, when sampling an image, we apply the same percent chance of varying the factor in each plot. This allows us to measure generalization when images from all instances have an equal chance of a factor changing. In Fig A14 on the other hand, we study how well models are able to generalize variation across instances. In this setup, a factor varies on for some instances (based on the proportion depicted in the graph). We hope the additional descriptions at the start of each section now clarify this distinction.

---

> > > ### Comment · Reviewer_AUUo · 2023-03-18
> > > **responses**
> > >
> > > Thanks for your responses.  The explanation of sampling procedures during training, and the gap measurement, are both much clearer.  I still wish a discussion around Kumar &al (mentioned in my initial review) were included as well; that remains absent.  Still, my largest concerns around better explanations and potential differences in training sample sizes have been addressed.
> > >
> > >
> > >
> > > Reviewer AUEY also asks an interesting question (5), which I think could be more concretely explained, beyond the statements in the new paper revision (though this is just one of many parts of the paper):
> > >
> > > “For example, while position and lighting have gaps of -5% and -2% respectively with no variability (Table 1), their gaps when 5% of instances vary are nearly -40% (Figure A14).”
> > >
> > > Some additional notes around this:
> > >
> > > First, it's important to be absolutely clear that this isn't due to imbalanced sampling.
> > >
> > > If 5% of instances are variable, would this affect the relative sampling proportions during training?  It's unclear what the sampling method is:  choose an instance and then transform (or not), so there are equal samples of each instance?  or add random transforms first, then sample, so there are many more samples for the varying instance?  I believe the procedure is the former, but this is still not fully clear to me.
> > >
> > > Second, assuming the above is resolved, it could be that the sudden jump down could be due to learning the variability as determining the object class, since now variation and class have been correlated.  For example, suppose "airplane" is varied by position, while all other classes use canonical position.  Then the center of the image now shows the background for airplane images, rather than the object --- thus any image where the center of the image is the background may now be classified as airplane.  If this is what is happening, it demonstrates how a classifier will fit to the features present, so that non-uniform application of transformations, rather than being generalized to all classes, are instead used as features to identify specific classes.  While that is a well-known phenomenon, this would be a stark demonstration of it if confirmed.

---

> > > > ### Author Response · Authors · 2023-04-05
> > > > **Response**
> > > >
> > > > We apologize for not including a more thorough discussion of Kumar & al. We have now included it and discussed its findings and its relationship with our work.
> > > >
> > > > You raise a good point: an imbalance in instance or class sampling are both critical to account for to avoid spurious correlations. We ensure both 1) our instance sampling is balanced in Sections 5.1-5.2 and, 2) classes are not disproportionately sampled thus introducing spurious correlations between factor changes and class labels in Section 5.4. We clarify the experimental setup and sampling procedure here. In Section 5.1, all instances are sampled in precisely the same manner. Specifically, we select a fixed probability for each setting (x-axis of Figure 4) indicating the probability with which to alter each factor across all instances. In Section 5.2, we study how well models generalize variation when only some instances vary. For this setting, the probabilities indicate the proportion of object instances for which each factor changes. Finally, in Section 5.4, we study how well models generalize across classes. For this setting, samples vary across factors for only a fixed set of half of the classes during training (Table 2). We’ve included more explicit descriptions of each setting in the camera ready submission to improve the clarity and thank you for pointing this out.

---

### Review · Reviewer_AUEY · 2023-01-22

**Summary Of Contributions:**

The paper analyzes the invariances of a range of modern image recognition models on a synthetic dataset showing rendered 3D objects against random 2D backgrounds. Factors of variation include object pose, position, size, color, as well as the background image. The paper shows that the models are not really robust to object transformations. Moreover, the paper shows that transfer of invariances across classes is difficult.

**Audience:**

Yes

**Broader Impact Concerns:**

No concerns, the paper studies robustness of models, which if anything could be a positive, contributing to our better understanding of the functioning of the models and potentially their biases.

**Claims And Evidence:**

No

**Requested Changes:**

Fix the issues pointed out in "weaknesses". in particular:
1. Add evaluation on ObjectNet and potentially human results (or otherwise explain the relevance of the proposed analysis)
2. Clarify the strange seemingly noisy results in the paper
3. Clarify the fine-tuning/linear protocols and evaluate their effect on the conclusions regarding robustness
4. (nice-to-have) Add more detailed results on performance depending on specific values of different transformations
5. Add missing reference(s) and polish writing, in particular related work w.r.t. Abbas & Deny
6. Add more models, in particular larger models like in Abbas & Deny

**Strengths And Weaknesses:**

Strengths:
1. A fairly thorough study, including a reasonable selection of modern vision models
2. The conclusion about limited robustness of modern models is interesting, the study of transfer of invariances across classes and instances is interesting too
3. It is interesting that MAE seems to behave substantially different from other models in terms of robustness

Weaknesses:
1. While I generally appreciate analysis papers, I wonder how exactly are the results on the proposed synthetic dataset relevant to real applications of image recognition models. It is no big surprise that the models do not work very well on out-of-distribution samples (like, say, upside-down objects on weird backgrounds), but do they even have to if such samples never appear in real applications? One argument could be that human visual recognition is invariant to all these variations, but then it’s necessary to explicitly point this out and support with experiments. Moreover, I would recommend to add evaluation on ObjectNet - a real-world dataset also concerned with measuring robustness of models - and see if conclusions there correlate with conclusions on the proposed synthetic dataset.
2. I am confused by the fact that in many plots the gap is not zero when I would expect it to be zero. For instance, in Figure 4 how come many curves don’t reach 0 on the right, at 100%? Perhaps because the highest evaluated percentage is 95% not 100%, but that’s weird and I’d like to see 100% too just to be sure. Another example is that in Figure 5 some curves go above 0 on the right, how can that be? These results make me doubt the validity of all conclusions, perhaps there is a lot of noise in the numbers?
3. From the numbers provided in the supplementary material, it seems the models overfit substantially during fine-tuning. It therefore seems that the specific fine-tuning protocol can have a large impact on the results and the conclusions of the experiments. However, I did not see a discussion of this in the paper.
4. It would be great to add results on which specifically poses, positions, sizes, colors, backgrounds are easy or difficult for the models. The synthetic nature of the proposed dataset should allow to generate such results? Would be interesting to see.
5. “For example, while position and lighting have gaps of -5% and -2% respectively with no variability (Table 1), their gaps when 5% of instances vary are nearly -40% (Figure A14).” - this is very strange, any explanation? Maybe training data imbalance during fine-tuning?
6. In comparison to a related paper by Abbas & Deny, the authors state that they analyze “a larger catalog of more recent SoTA models and architectures” - does not seem quite accurate: while some of the models studied in this paper are not in Abbas & Deny, the opposite is also true. Afaict, Abbas & Deny study 38 models, while this submission - 11 models, is it right?  Then the statement about “larger catalog” does not seem quite correct.
    6a. In particular, Abbas & Deny show that large models and in particular EfficientNets trained with Noisy Student are quite a bit more robust than other models, it would be very valuable check this on the larger dataset proposed in this submission
7. Smaller issues
    7a. Missing reference “On Robustness and Transferability of Convolutional Neural Networks” (Djolonga et al.)
    7b. It seems confusing that for a generally poorly performing  model the gap is small and therefore it seems that the model is more invariant. Perhaps relative gap would be a better metric?

---

> ### Author Response · Authors · 2023-02-17
> **Responses**
>
> We thank you for such helpful detailed feedback on our work. We appreciate that you found our study to be thorough and found our conclusions interesting from models’ inability to generalize variation to particularities of MAE. Thanks to your feedback we’ve made several improvements to our work including a new analysis of two new models (BiT and EfficientNetL2 trained on JFT-300M) in Appendix F, a new analysis of models sensitivity to factor parameters (Appendix E), additional tables with both absolute accuracies and relative gaps in Appendix C as well as several clarifications in the text to ensure our claims are lucid and well-supported. We discuss in detail specific points below:
>
> > 1. human visual recognition is invariant to all these variations, but then it’s necessary to explicitly point this out and support with experiments.
>
> We thank you for the suggestion. We have provided the range of variation for the factors of variations for various (randomly selected) objects in supplementary Figures A1 to A10. These images make it clear that humans are able to recognize objects when these transformations are applied and as such the significant drops in performance observed with neural networks are not observed with humans. While we agree ObjectNet is an interesting evaluation dataset, it does not allow us to study generalization of factors between training and testing—this is the core contribution of our work. Furthermore, for many of the models we study, out-of-the-box evaluation of ObjectNet is already available in prior work. For example, the original CLIP paper [2] already evaluates ObjectNet classification performance.
>
> We appreciate the suggestion to include motivation by connecting robustness to human/animal visual systems. Prior work has studied the identifiability of objects under various rotations for primates [1], and the conclusions were that for easily identifiable objects (such as a teapot), primate vision is extremely robust and the objects need to be very artificial (tangled wires) to make this task difficult. We now include a discussion of this motivation regarding primates in the text (marked in blue). This contrasts with the failure modes we observe for neural networks, where changes that would not lead to any drop in performance for humans or primates do lead to a drop in performance. We provide more details below when giving a more detailed analysis of the failure modes under pose variations.
>
> [1] Logothetis, N. K., et al. "View-dependent object recognition by monkeys." Current biology 4.5 (1994): 401-414.
> [2] Radford, Alec, Jong Wook Kim, Chris Hallacy, Aditya Ramesh, Gabriel Goh, Sandhini Agarwal, Girish Sastry, et al. “Learning Transferable Visual Models From Natural Language Supervision.” https://doi.org/10.48550/arXiv.2103.00020.
>
>
> > 2. “I am confused by the fact that in many plots the gap is not zero when I would expect it to be zero. For instance, in Figure 4 how come many curves don’t reach 0 on the right, at 100%?”
>
> It’s absolutely possible for curves of the performance gaps to not reach 0 and it’s also absolutely possible for gap curves to also reach above 0. To clarify the setup of the experiments, in Figure 4 we measure how well models generalize changes in each factor when training samples contain the same changes. Each point along the x-axis indicates the chance a training sample would contain a randomly varying pose, position, and so on. You’re correct: the x-axis maximum is 95% not 100% (as described in Section 5.1). On the y-axis we plot for each training setting the corresponding performance gap measured as the accuracy on held-out samples randomly varying for the given factor - accuracy on held-out samples in their canonical setting. In Figure 5, the values above zero for position indicate the model is on average more adept at recognizing samples randomly varying in position compared to samples in the canonical position. This can occur when during training the chance of instances varying poses is greater than the canonical pose. For example, when the “percent of varying training instances” (x-axis) is 95%, only 5% of instances are seen in their canonical position while the other 95% are seen with position randomly varying. To improve clarity, we now detail each experimental setting under the caption of each figure (indicated in blue).

---

> > ### Author Response · Authors · 2023-02-17
> > **Responses continued (2)**
> >
> >
> > >  3. "From the numbers provided in the supplementary material, it seems the models overfit substantially during fine-tuning. It therefore seems that the specific fine-tuning protocol can have a large impact on the results and the conclusions of the experiments. However, I did not see a discussion of this in the paper."
> >
> > We thank the reviewer for pointing out we don’t explicitly discuss the potential for overfitting for fine-tuning—in particular when variation in training is only present for some instance/objects. When few instances vary during training, it's easier for models to overfit rather than generalize. We’ve corrected this by including a discussion (highlightedd in blue on pg 11) in the revised draft. We also note in every experiment, we analyze every model with both linear evaluation and finetuning protocols.
> >
> > > 4. “It would be great to add results on which specifically poses, positions, sizes, colors, backgrounds are easy or difficult for the models. The synthetic nature of the proposed dataset should allow to generate such results? Would be interesting to see.”
> >
> > We thank the reviewer for their suggestion. We agree this is a very interesting study and we added it to Appendix Section E (and Figure A12). As we explain in the paper which we edited, we perform the analysis for the best performing model, iBot trained on ImageNet-21k, according to held-out accuracy. We find Pose and Scale have the highest accuracy for the canonical value (0.0 and 1.0 respectively)  with accuracy dropping as the variation strength increases and for position the drop is more muted compared to that for pose and scale. For Lighting color, we do not observe significant changes in accuracy as Lighting color strength varies, but this is aligned with the smaller performance gaps we observe for lighting color.
> >
> > Going back to our reply for Point 1 and focusing on pose, we see similar behavior for a relatively simple task (classification) for neural networks as is observed in primates for a more complex one (identifiability). As such, even though neural networks are less robust since they fail on simpler tasks, they exhibit similar behaviors as primates when they do fail.
> >
> > > 5. “For example, while position and lighting have gaps of -5% and -2% respectively with no variability (Table 1), their gaps when 5% of instances vary are nearly -40% (Figure A14).”
> >
> > Indeed, we note that adding variability only for some instances (or classes) but not all leads to worse generalization, as we mention in our original submission:
> >
> > “Interestingly, varying only a portion of instances led to substantial overfitting, especially when the proportion of varying instances is smaller than 50% [...] This suggests models struggle to generalize variation across instances so much so that it can hurt generalization relative to seeing no variability.”
> >
> > To better explain what is happening, we modified the text in this section to emphasize this point:
> >
> > “This suggests that when the specific subset of training instances seen varying during learning is small, the model overfits to this small set of varying instances. In comparison with when no variability was seen at all during learning (Table 1), the model’s generalization gap, i.e. its ability to recognize test instances under varying factors values relative to its performance on canonical test instances, is increased.”
> >
> > > 6. In comparison to a related paper by Abbas & Deny, the authors state that they analyze “a larger catalog of more recent SoTA models and architectures” - does not seem quite accurate: while some of the models studied in this paper are not in Abbas & Deny, the opposite is also true. Afaict, Abbas & Deny study 38 models, while this submission - 11 models, is it right? Then the statement about “larger catalog” does not seem quite correct. In particular, Abbas & Deny show that large models and in particular EfficientNets trained with Noisy Student are quite a bit more robust than other models, it would be very valuable check this on the larger dataset proposed in this submission
> >
> > We apologize for the poor phrasing in comparing the catalog of models to prior work. We thank you for pointing this out. We now include an analysis of both EfficientNet and BiT (trained on JFT-300) and have rephrased our comparison to more accurately reflect the range of models we study in relation to prior work. We note that in addition to these new models, our submission does include 4 models trained on ImageNet-21k (13 million images) as well as two CLIP models trained on (400 million and 2 billion text-image pairs).

---

> > > ### Author Response · Authors · 2023-02-17
> > > **Responses continued (3)**
> > >
> > >
> > > > 7a. “Missing reference “On Robustness and Transferability of Convolutional Neural Networks” (Djolonga et al.)“
> > >
> > > We thank the reviewer for pointing out this related work. We have added it to our related work section and explained our difference with their work as follows:
> > >
> > > Similar to us, Djolonga et al. (2020) studies robustness of models with respect to factors of variation of objects on various backgrounds, and how inductive biases such as the amount of data or size of model impact performance. However, they only focus on Convolutional Neural Networks (CNNs), while we also study vision transformers based architectures, feedforward models and even zero-shot multi-modal models such as CLIP. Furthermore, while they study the factors orientation (rotation), position and size, their rotations are only changing 2D in-plan orientation, while we use realistic 3D renderings of objects. More importantly, we extend our study to the case where variability in each factor is added \emph{during training}, and not just at test-time. This allows us to gain an in-depth understanding of models ability to generalize seen variations to new objects, rather than out-of-the-box robustness.
> > >
> > > > 7b. “It seems confusing that for a generally poorly performing model the gap is small and therefore it seems that the model is more invariant. Perhaps relative gap would be a better metric?”
> > >
> > > We thank the reviewer for their suggestions and have added relative numbers in Table A4 and A5 for both linear evaluation and finetuning protocols.
> > >
> > >
> > > We hope the detailed responses above along with the experiments on new models, new sensitivity analysis, additional tables of results, and clarifications in the text address your concerns. We remain at your disposal to answer any further questions.

---

### Review · Reviewer_WbwZ · 2023-02-03

**Summary Of Contributions:**

The work evaluates common vision models (image classification) including novel architectures (CLIP, SSL) wrt. their robustness under variation of natural visual factors (object pose, position, scale, color, background). The work highlights a category of significant generalization gaps for all current approaches, showing poor generalization to out-of-distribution cases. The work also evaluates other interesting generalization aspects, such as generalization across classes, under different amounts of factor variation, with / without fine-tuning.

**Audience:**

Yes

**Broader Impact Concerns:**

The work sheds a light on the capabilities and in-capabilities of current vision models and emphasizes the need for more work on model generalization and data biases.

**Claims And Evidence:**

Yes

**Requested Changes:**

I think the paper is already in a very good state and I would only have some minor change requests. Please see the minor weaknesses above.

**Strengths And Weaknesses:**

Strengths:

The paper addresses the important topic of robustness / generalization of current vision architectures to natural image variations. It provides strong empirical evidence that current architectures lack generalization wrt certain visual factors. The work is highly relevant for the CV community and raises concerns in using current vision architectures in safety critical use-cases. The paper is well written and structured.


Weakness (minor):

* In some figure's notations are rather small and should be enlarged to improve accessibility (Fig 2, 3, 4, 5, 6, 7)

* The following reference addressing a similar question on largely the same visual factors should be added / discussed:

DiagViB-6: A Diagnostic Benchmark Suite for Vision Models in the Presence of
Shortcut and Generalization Opportunities, https://openaccess.thecvf.com/content/ICCV2021/papers/Eulig_DiagViB-6_A_Diagnostic_Benchmark_Suite_for_Vision_Models_in_the_ICCV_2021_paper.pdf

---

> ### Author Response · Authors · 2023-02-17
> **Thank you for the thoughtful feedback**
>
> We appreciate your thoughtful feedback and agree a systematic study of current vision models’ generalization would be highly relevant for the research community. We thank you for pointing out larger notations would improve accessibility. We’ve updated Figures 2, 3, 4, 5, 6, and 7 with larger notations. We apologize for our oversight and now include a discussion of DiagViB-6 in our related work.

---

### Author Response · Authors · 2023-02-17
**Response**

We appreciate the reviewers’ careful consideration of our work and thoughtful feedback. We’re glad all reviewers found the topic of robust generalization “important” and “highly relevant” (Wbwz) to the vision research community. We’re also glad reviewers found our study “thorough” (AUEY and AUUO) and conclusions regarding models’ ability to generalize changes in natural factor seen during training “interesting” (AUUo) and lead to a “better understanding of the functioning of the models and potentially their biases” (AUEY).

Thanks to reviewers’ feedback, we’ve made several improvements to our submission including a new analysis of two new models, a new analysis of models' sensitivity to factor parameters, additional tables to summarize findings well as several clarifications in the text (highlighted in blue) to ensure our claims are lucid and well-supported. We believe the revised submission would be highly valuable to the research community.

We remain available to incorporate any further suggestions.

---

### Decision · Action_Editors · 2023-03-21

**Recommendation:** Accept as is

**Comment:**

I believe this is important kind of work, since it furthers our understanding about the limitations of these models. I would encourage the authors to include some of the unresolved discussions mentioned by the reviewers (e.g. Kumar et al) in the camera ready manuscript, but also to shorten the main text to the page limit.

**Audience:**

The audience of this work is anyone who is building computer vision models and wants to understand the robustness of the methods to various factors of variation.

**Claims And Evidence:**

This works looks into invariances of modern image recognition models on synthetic data. Lots of factors of variation are considered and the results are that models are not actually that robust to these transformations. The work also looks into other kinds of generalization, e.g. across classes.

The reviewers noted that this is a strong empirical work that provides ample evidence for what kind of things current architectures are not robust to (from a visual point of view). While all three reviewers pointed out a number of weaknesses in the study, the authors seemed to have addressed most of them satisfactorily in the rebuttal. The updated version seems to have a number of extra experiments to confirm the validity of the conclusions.